# Multi-tiered actions of *Legionella* effectors to modulate host Rab10 dynamics

**Tomoko Kubori[1]\*, Kohei Arasaki[2], Hiromu Oide[2], Tomoe Kitao[1], Hiroki Nagai[1,3]\***

[1]Department of Microbiology, Graduate School of Medicine, Gifu University, Gifu, Japan; [2]School of Life Sciences, Tokyo University of Pharmacy and Life Sciences, Hachioji, Japan; [3]Center for One Medicine Innovative Translational Research (COMIT), Gifu University, Gifu, Japan

**\*For correspondence:**
kubori.tomoko.j0@f.gifu-u.ac.jp (TK);
nagai.hiroki.d3@f.gifu-u.ac.jp (HN)

**Competing interest:** The authors declare that no competing interests exist.

**Abstract** Rab GTPases are representative targets of manipulation by intracellular bacterial pathogens for hijacking membrane trafficking. *Legionella pneumophila* recruits many Rab GTPases to its vacuole and exploits their activities. Here, we found that infection-associated regulation of Rab10 dynamics involves ubiquitin signaling cascades mediated by the SidE and SidC families of *Legionella* ubiquitin ligases. Phosphoribosyl-ubiquitination of Rab10 catalyzed by the SidE ligases is crucial for its recruitment to the bacterial vacuole. SdcB, the previously uncharacterized SidC-family effector, resides on the vacuole and contributes to retention of Rab10 at the late stages of infection. We further identified MavC as a negative regulator of SdcB. By the transglutaminase activity, MavC crosslinks ubiquitin to SdcB and suppresses its function, resulting in elimination of Rab10 from the vacuole. These results demonstrate that the orchestrated actions of many *L. pneumophila* effectors fine-tune the dynamics of Rab10 during infection.

## eLife assessment

This **important** study explores the interplay between Legionella Dot/Icm effectors that modulate ubiquitination of the host GTPase Rab10, which undergoes phosphoribosyl-ubiquitination by the SidE family of effectors, which in turn are required for Rab10 recruitment to the Legionella containing vacuole (LCV). The evidence supporting the claims of the authors is **convincing**. The study is not only relevant for the microbiology community, but will also be of interest to colleagues in the broader fields of membrane trafficking and general cell biology.

## Introduction

*Legionella pneumophila* utilizes a large arsenal of effector proteins which are delivered via its Dot/Icm type IV secretion system (T4SS) to modulate host cellular systems (*Hubber and Roy, 2010*; *Kubori and Nagai, 2016*). The functions of the effector proteins are crucial for establishing a replicative niche where *L. pneumophila* can survive and avoid host defense mechanisms in the cell (*Isberg et al., 2009*; *Qiu and Luo, 2017*). Many effector proteins are known to modulate the function of host Rab GTPases that regulates cellular membrane transport (*Neunuebel and Machner, 2012*; *Sherwood and Roy, 2013*; *Spanò and Galán, 2018*). Among them, effector manipulation of Rab1, a critical regulator of membrane trafficking between endoplasmic reticulum (ER) and the Golgi complex, has been extensively analyzed in *L. pneumophila* infection (*Arasaki et al., 2012*; *Ingmundson et al., 2007*; *Machner and Isberg, 2006*; *Mukherjee et al., 2011*; *Müller et al., 2010*; *Murata et al., 2006*; *Neunuebel et al., 2011*; *Tan and Luo, 2011*).

Ubiquitination regulates all aspects of eukaryotic cell physiology, and therefore is exploited by various bacterial pathogens encoding ubiquitin (Ub) ligases and deubiquitinases (DUBs) (*Ashida and Sasakawa, 2017*; *Kitao et al., 2020*; *Kubori et al., 2019*; *Lin and Machner, 2017*). Effector-mediated Ub modification of Rab1 was reported as an action of the *Legionella* E3 ligase SidC and its paralog SdcA (*Horenkamp et al., 2014*). Rab33b as well as Rab1, Rab6A, and Rab30 are conjugated with phosphoribosylated (PR)-Ub by the unique reaction mechanism of the SidE family of effector proteins (*Qiu et al., 2016*), and it was shown that recruitment of Rab33b and Rab6A to the *Legionella*-containing vacuole (LCV) relies on the PR ubiquitination of Rab33b (*Kawabata et al., 2021*). A recent genome-wide screen identified host factors including Rab10 linked to intracellular replication of *L. pneumophila* (*Jeng et al., 2019*). It was demonstrated that Rab10 is recruited to the LCV and is ubiquitinated depending on the activity of SidC and SdcA (*Jeng et al., 2019*; *Liu et al., 2020*).

SidC and SdcA were originally identified as tethering factors that function in ER-to-LCV trafficking both having a unique phosphatidylinositol-4 phosphate (PI[4]P)-binding domain (*Ragaz et al., 2008*). The enzymatic activity of these proteins as Ub ligases was experimentally uncovered later (*Hsu et al., 2014*). The enzymatic activity was found to be encoded in the domain conserved between the two proteins, namely SidC N-terminal Ub ligase (SNL) domain (*Hsu et al., 2014*). SdcB (Lpg2452/LegA14) was identified as another *L. pneumophila* Ub ligase having the SNL domain (*Lin et al., 2018*). However, the role of SdcB in *L. pneumophila* infection has not been examined yet.

Various *L. pneumophila* effector proteins exploit the host Ub signaling cascade (*Kitao et al., 2020*; *Luo et al., 2021*; *Tomaskovic et al., 2022*). They are exemplified by MavC which was shown to chemically modify the E2 enzyme UBE2N. The transglutaminase activity of MavC catalyzes a covalent linkage between Ub and UBE2N, thereby abolishing the activity of UBE2N to form polyUb chains, and mediate host NFκB signaling (*Gan et al., 2019a*). Encoded by the neighboring gene of *mavC*, MvcA is a paralog of MavC and exhibits similar activity to MavC as a Ub-specific deamidase (*Valleau et al., 2018*). However, MvcA was found to have an ability to reverse MavC-mediated Ub conjugation to UBE2N by its unique DUB activity (*Gan et al., 2020*). The enzymatic activities of MavC and MvcA are both inhibited by Lpg2149, an effector encoded downstream of *mvcA*, by blocking their catalytic residues (*Valleau et al., 2018*).

In this study, we investigated the role of *Legionella* Ub ligases in Rab10 dynamics during *L. pneumophila* infection. Including the unexpected finding that MavC is a negative regulator of SdcB, we demonstrated multi-tiered regulation by many effectors to finely modulate the localization of Rab10 to the LCV, revealing the intricate effector network that hijacks cellular processes.

## Results

### The SidE- and SidC-family proteins differentially contribute to ubiquitination of Rab10 in infected cells

Host Rab10 is required for optimal intracellular replication of *L. pneumophila* (*Jeng et al., 2019*) and therefore considered to play a significant role in LCV biogenesis or maintenance. Since *L. pneumophila* SidE-family proteins, which catalyze PR-linked ubiquitination, have a wide range of substrates including Rab1 and Rab33b (*Bhogaraju et al., 2016*; *Kalayil et al., 2018*; *Kotewicz et al., 2017*; *Qiu et al., 2016*), we first examined whether the SidE family can affect Rab10 ubiquitination. Upon infection of HEK293T-FcγRII cells transiently expressing FLAG-Rab10 and HA-Ub with a wild-type *L. pneumophila* strain (Lp01) for 1 hr, Rab10 was detected with a shifted band of higher molecular mass (*Figure 1a*, upper panel). The band was shown to contain Ub by probing with an anti-HA antibody (*Figure 1a*, lower panel), indicating that the band represents Rab10 conjugated with a single Ub molecule. The mass shift was not detected in cells infected with the T4SS-deficient strain (Δ*dotA*), suggesting that Rab10 can be monoubiquitinated in a T4SS-dependent manner. Infection with a Δ*sidEs* strain lacking all four SidE-family proteins (SidE, SdeA, SdeB, and SdeC) mostly eliminated the molecular mass shift of Rab10, while infection with a strain lacking the negative regulators of SidE-family proteins (DupA, DupB, SidJ, and SdjA; *Black et al., 2019*; *Gan et al., 2019b*; *Qiu et al., 2017*; *Shin et al., 2020*; *Sulpizio et al., 2019*) enhanced the intensity of the band (*Figure 1a*). These results suggest that the SidE-family proteins can conjugate PR-Ub to Rab10. The high molecular weight smears detected with anti-HA antibody are thought to be polyUb chains (*Figure 1a*, lower panel). Appearance of these

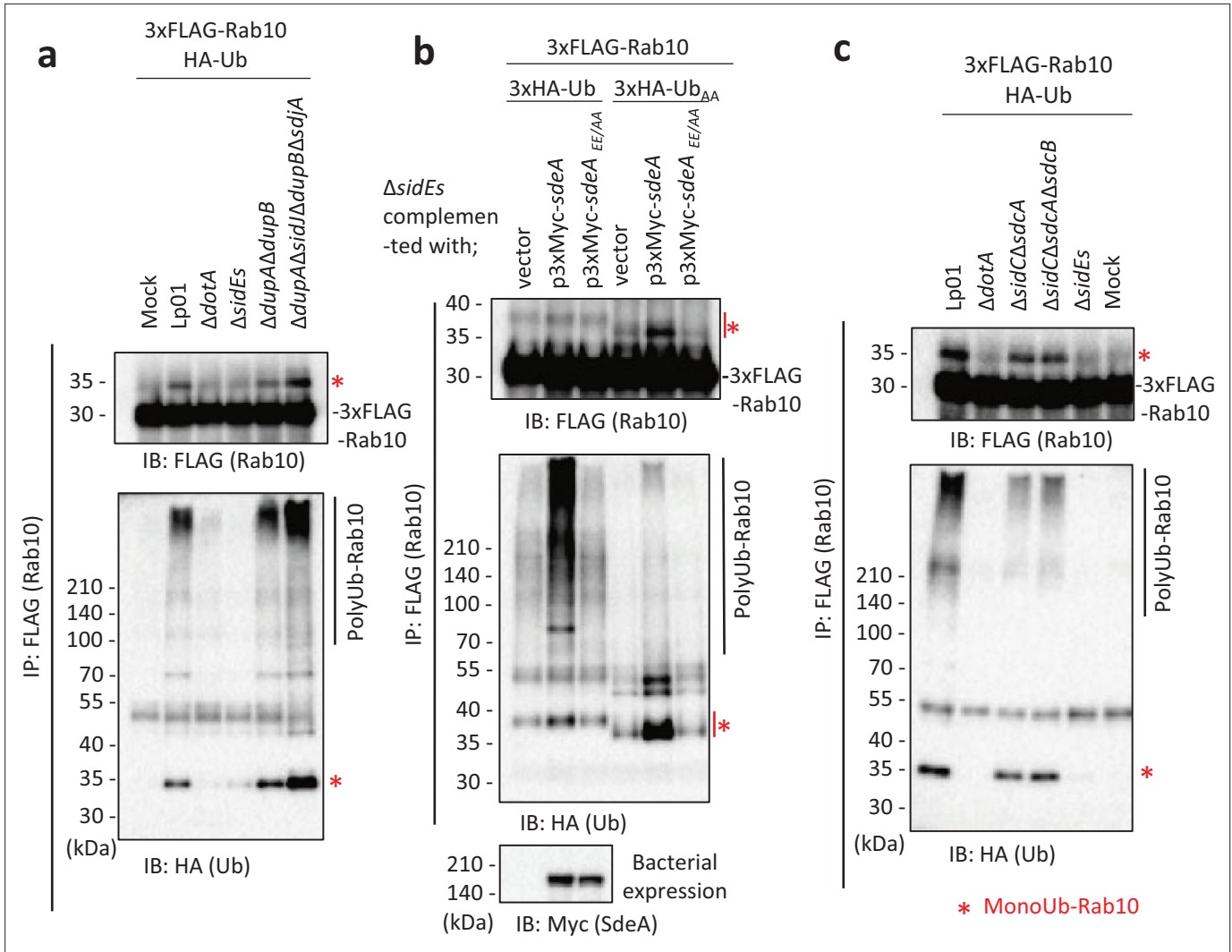

**Figure 1.** The SidE- and SidC-family proteins differentially contribute toward ubiquitination of Rab10. HEK293T-FcγRII cells transiently expressing 3xFLAG-Rab10 and HA-Ub were infected with the indicated *L. pneumophila* strains for 1 hr at a multiplicity of infection (MOI) of 20. Rab10 was isolated from cell lysate by immunoprecipitation using anti-FLAG magnetic beads and was probed with anti-FLAG and with anti-HA antibodies (**a-c**). Triple-HA-Ub (3xHA-Ub) or Ub in which the C-terminal GG were replaced with AA (3xHA-Ub$_{AA}$) was expressed instead of HA-Ub in (**b**). Bacterial lysates were probed with anti-Myc antibody in (**b**). The asterisks indicate the postion of monoubiquitinated Rab10.

The online version of this article includes the following source data and figure supplement(s) for figure 1:

**Source data 1.** Original files for the western blot analysis in *Figure 1a* (anti-FLAG and anti-HA).

**Source data 2.** PDF containing *Figure 1a* and original scans of the relevant western blot analysis (anti-FLAG and anti-HA), with cropped areas.

**Source data 3.** Original files for the western blot analysis in *Figure 1b* (anti-FLAG, anti-HA, and anti-Myc).

**Source data 4.** PDF containing *Figure 1b* and original scans of the relevant western blot analysis (anti-FLAG, anti-HA, and anti-Myc), with cropped areas.

**Source data 5.** Original files for the western blot analysis in *Figure 1c* (anti-FLAG and anti-HA).

**Source data 6.** PDF containing *Figure 1c* and original scans of the relevant western blot analysis (anti-FLAG and anti-HA), with cropped areas.

**Figure supplement 1.** Mutation of Lys102, Lys136, and Lys154 on Rab10 did not eliminate ubiquitination of Rab10 upon infection.

**Figure supplement 1—source data 1.** Original files for the western blot analysis in *Figure 1—figure supplement 1a* (anti-FLAG and anti-HA).

**Figure supplement 1—source data 2.** PDF containing *Figure 1—figure supplement 1a* and original scans of the relevant western blot analysis (anti-FLAG and anti-HA), with cropped areas.

**Figure supplement 1—source data 3.** Original files for the western blot analysis in *Figure 1—figure supplement 1b* (anti-FLAG and anti-HA).

**Figure supplement 1—source data 4.** PDF containing *Figure 1—figure supplement 1b* and original scans of the relevant western blot analysis (anti-FLAG and anti-HA), with cropped areas.

smears is tightly associated with the PR-Ub bands, showing that polyubiquitination of Rab10 is linked with its PR ubiquitination.

To examine if PR-Ub conjugation to Rab10 can enhance its modification with polyUb chains, we infected HEK293T-FcγRII cells expressing FLAG-Rab10 and HA-Ub with the ΔsidEs strain complemented with a plasmid expressing wild-type or a catalytic mutant of SdeA (SdeA$_{EE/AA}$), a representative SidE-family protein (*Figure 1b*). We detected a strongly enhanced polyUb smear whose appearance depends on the mono-ADP ribosyltransferase activity of SdeA. The intensity of the monoubiquitination band was correlated with that of the polyUb smears (*Figure 1b*, middle panel). When the C-terminal GG motif of Ub was replaced with AA (Ub$_{AA}$), the polyUb smear drastically diminished, and accumulation of monoUb-conjugated Rab10 was observed instead. This indicates that the polyUb chains on Rab10 were formed via the Ub C-terminus by the canonical ubiquitination reaction. This also shows that the observed monoUb conjugation to Rab10 does not require the Ub C-terminus, which is consistent with the formation of bridge between Arg42 of Ub and substrate catalyzed by the SidE effectors (*Bhogaraju et al., 2016*; *Kotewicz et al., 2017*; *Qiu et al., 2016*). These results strongly support that Rab10 is subjected to SdeA-mediated PR ubiquitination and that this modification may provide a platform of conjugation of polyUb chains to Rab10, which is expected to be mediated by other Ub ligases.

Since it was reported that *L. pneumophila* effectors SidC and its paralog SdcA induce Rab10 ubiquitination (*Jeng et al., 2019*; *Liu et al., 2020*), we investigated how these E3 ligases contribute toward ubiquitination of Rab10 in our system (*Figure 1c*). Infection of cells with a ΔsidCΔsdcA strain as well as with a strain lacking all the SidC-family proteins (ΔsidCΔsdcAΔsdcB) still caused both mono- and polyubiquitination of Rab10 but with reduced levels. On the contrary, infection with the ΔsidEs strain eliminated both modifications. These results support that SidE-family proteins primarily contribute toward ubiquitination of Rab10, and that SidC-family proteins partly contribute toward polyubiquitination of Rab10 directly or indirectly in conditions where Rab10 is modified with PR-Ub. We also found that substitution of three residues on Rab10 (Lys102, Lys136, and Lys154) that were previously suggested to be potential ubiquitination sites (*Jeng et al., 2019*) to Ala (Rab10KKK) did not abolish polyubiquitination of Rab10 (*Figure 1—figure supplement 1a*), implying that ubiquitination sites can be present in the other residues. This raises the possibility that canonical Ub chains can be formed partly on PR-Ub conjugated to Rab10. The polyubiquitination of Rab10 was significantly observed until 7 hr after infection, although the level was slightly reduced (*Figure 1—figure supplement 1b*).

## Rab10 recruitment to the LCV is differentially regulated by SidE- and SidC-family proteins

Earlier studies demonstrated that Rab10 ubiquitination is highly correlated with its localization to the LCV (*Jeng et al., 2019*; *Liu et al., 2020*). We therefore examined if the SidE family regulates Rab10 recruitment to the LCV using HeLa-FcγRII cells transiently expressing RFP-Rab10 (*Figure 2a and b*). Infection with the ΔsidEs strain drastically reduced the level of Rab10-positive LCVs at all time points examined (*Figure 2b*). As reported for Rab33b (*Kawabata et al., 2021*), PR ubiquitination is thought to be required for Rab10 to localize to the LCV. The ΔsidCΔsdcA LCV also exhibited a reduced the level of Rab10 localization at 1 hr after infection (*Figure 2b*, left panel). However, the level of Rab10 recruitment to the ΔsidCΔsdcA LCV recovered at 4 hr after infection, while that to the ΔsidCΔsdcAΔsdcB LCV did not recover even as late as 7 hr (*Figure 2b*, middle and right panels). This result suggests that SdcB can contribute toward retention of Rab10 on the LCV at late stages of infection.

## SdcB associates with the LCV and plays a major role in Ub recruitment to the LCV at late stages of infection

The Ub accumulation on the LCV has been thought to be mediated largely by SidC and SdcA (*Horenkamp et al., 2014*; *Luo et al., 2015*). We therefore examined the possible role of SdcB in Ub recruitment to the LCV at distinct time points after infection (*Figure 3*). To mask the effect of SidC and SdcA, we used a ΔsidCΔsdcAΔsdcB strain complemented with wild-type or catalytic mutant (C57A) of SdcB expressed from a plasmid. When HeLa-FcγRII cells were infected with these strains for 1 hr, SdcB was detected around the LCV (*Figure 3a, b*). The level of wild-type SdcB-positive LCVs was relatively lower compared with that of the SdcB$_{C57A}$-positive LCVs, probably due to its catalytic cycling. The expression of wild-type SdcB led to recruitment of Ub to the vacuole even without SidC and SdcA,

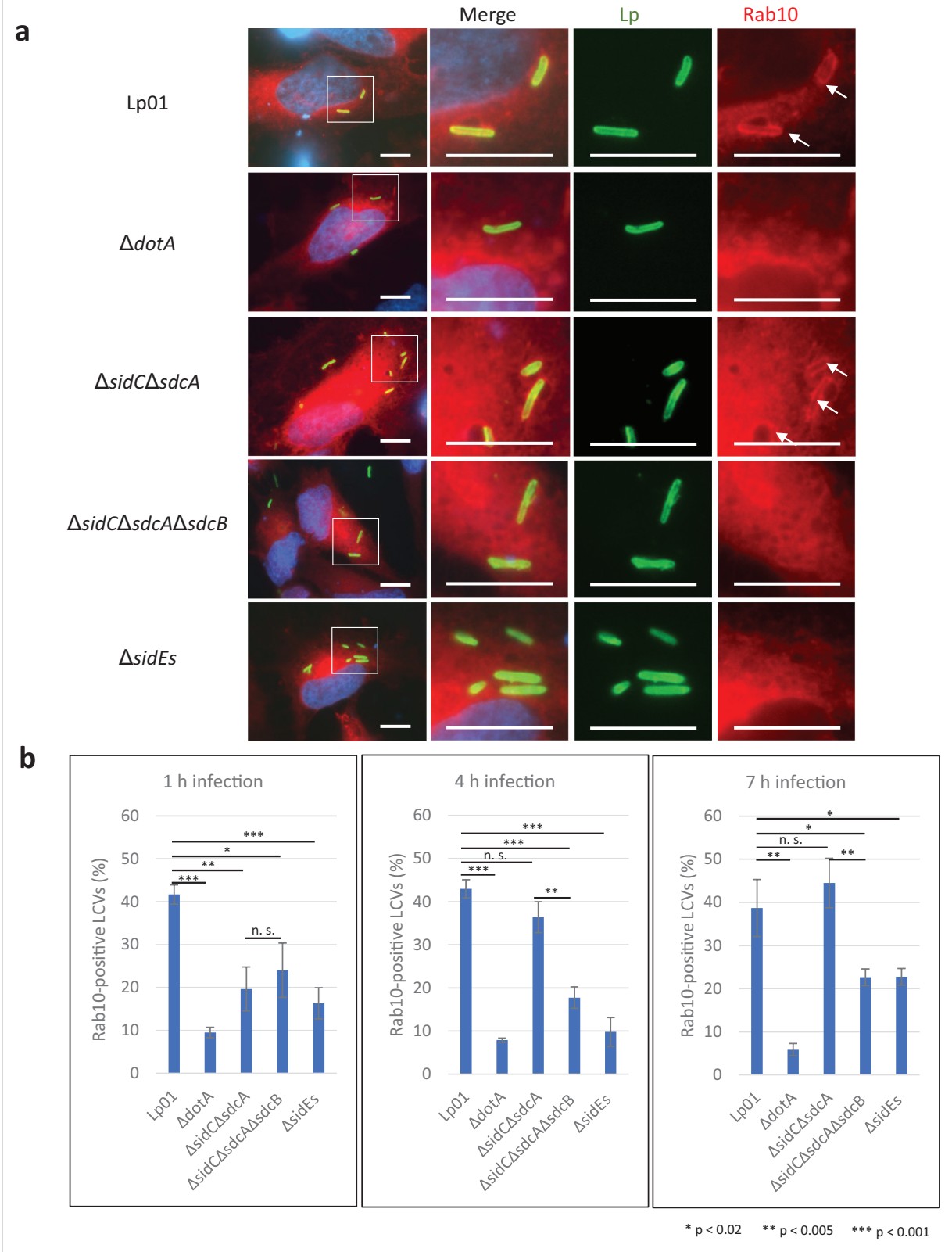

**Figure 2.** The SidE- and SidC-family proteins differentially contribute toward recruitment of Rab10 to the *Legionella*-containing vacuole (LCV). HeLa-FcγRII cells transiently expressing RFP-Rab10 were infected with the indicated *L. pneumophila* strains at an MOI of 5 for 4 hr (**a**) and for the indicated time (**b**). (**a**) Representative images of infected cells. Fixed cells were stained for *L. pneumophila* (green) and DNA (blue) and visualized with RFP-Rab10 (red). Magnified images in the white squares are shown in merged and in each channel. Arrows indicate the Rab10-positive LCVs. Scale bars, 10 μm.

*Figure 2 continued on next page*

Figure 2 continued

(**b**) Quantitation of Rab10-positive LCVs (%). Infections were performed in triplicate and each value represents scoring from 200 LCVs. Significance was determined using Student's *t*-test.

The online version of this article includes the following source data for figure 2:

**Source data 1.** Raw images of micrographs in *Figure 2a*.

**Source data 2.** Counting data in *Figure 2b*.

while expression of SdcB$_{C57A}$ did not (*Figure 3b*). This result indicates that SdcB has a catalytic activity to conjugate Ub to substrates on the LCV. At 7 hr after infection, the level of the SdcB-positive LCVs was elevated to about 70% regardless of its catalytic activity (*Figure 3c, d*). At this time point, the level of the Ub-positive LCVs was also raised to about 60 %, but only when SdcB was catalytically active. These results demonstrate that SdcB can play a substantial role to conjugate Ub to substrates on the LCV at late stages of infection.

## The catalytic activity of SdcB enhances retention of Rab10 on the LCV

As Rab10 is one of the plausible substrates located on the LCV, we assessed its ubiquitination upon infection with the *L. pneumophila* strains for comparison between its GTP-bound active (QL) and GDP-bound inactive (TN) conformations (*Figure 4a*). A substantial amount of ubiquitinated Rab10QL, but not Rab10TN, was detected upon infection with the wild-type strain (*Figure 4a*; *Figure 4—figure supplement 1*). We found that ubiquitination of Rab10TN detected with HA-probing (HA-Ub) was apparently indirect and not infection-induced, as FLAG-probing (FLAG-Rab10) mostly eliminated the high molecular weight smears on Rab10TN (*Figure 4—figure supplement 1*). Polyubiquitinated Rab10QL was reduced by infection with the Δ*sidC*Δ*sdcA*Δ*sdcB* strain, while both mono- and poly-ubiquitination of Rab10QL was mostly blocked by infection with the Δ*sidEs* strain. Localization of active Rab10 (Rab10QL) on the LCVs was apparently correlated with the level of its polyubiquitination (*Figure 4b*, *Figure 4—figure supplement 2*). The finding that the active form of Rab10 is preferentially targeted and spatially regulated by the action of SidE- and SidC-family ligases prompted us to examine the effect of SdcB activity on ubiquitination of this specific form of Rab10. Bacterially expressed SdcB, but not its catalytic mutant, significantly enhanced the level of polyubiquitination of Rab10QL at 7 hr after infection (*Figure 4c*). To examine the relationship between the catalytic activity of SdcB and the LCV localization of Rab10, we assessed the level of Rab10-positive LCVs on which SdcB localized. At 7 hr post infection, Rab10 localization was readily detected on Δ*sidC*Δ*sdcA*Δ*sdcB* LCVs containing wild-type SdcB, but not the inactive SdcB$_{C57A}$ mutant (*Figure 4d*). The level of Rab10-positive LCVs was significantly higher with expression of wild-type SdcB than that of the catalytic mutant (*Figure 4e*), suggesting that the Ub ligation activity of SdcB supports retention of Rab10 on the LCV until late stages of infection.

## MavC modifies SdcB when ectopically expressed in cells

A potential relationship between SdcB and MavC as an effector/metaeffector pair was suggested by a recent systematic analysis utilizing yeast genetics (*Urbanus et al., 2016*). We therefore examined whether expression of MavC can affect the activity profile of SdcB in cells. When ectopically expressed in HEK293T-FcγRII cells, 3xFLAG-tagged SdcB could not be detected (*Figure 5a*, top panel, most left lane). However, we found that coexpression of GFP-tagged MavC, but not of its catalytic mutant (MavC$_{C74A}$), recovered the detection of SdcB (*Figure 5a*, top panel). As SdcB was resolved as a doublet in the immunoblot, we suspected that SdcB may be chemically modified by MavC and that the modification may result in enhanced detection of this protein. We therefore probed with an anti-HA antibody to detect possible Ub conjugation. The upper band was stained with anti-HA antibody, showing that Ub was conjugated to SdcB presumably by the known Ub conjugation ability of MavC (*Gan et al., 2019a*; *Figure 5a*, middle panel). We also found that the disappearance of the SdcB bands correlated with the appearance of the high molecular weight smears when probed with anti-HA antibody (*Figure 5a*, middle panel). This suggests that the disappearance of the SdcB band can be caused by auto-ubiquitination, as SdcB has an ability to catalyze auto-ubiquitination with a diverse repertoire of E2 enzymes (*Figure 5—figure supplement 1*) consistently with a previous report (*Lin et al., 2018*). A proteasome inhibitor MG132 treatment of the cells did not rescue the disappearance of the

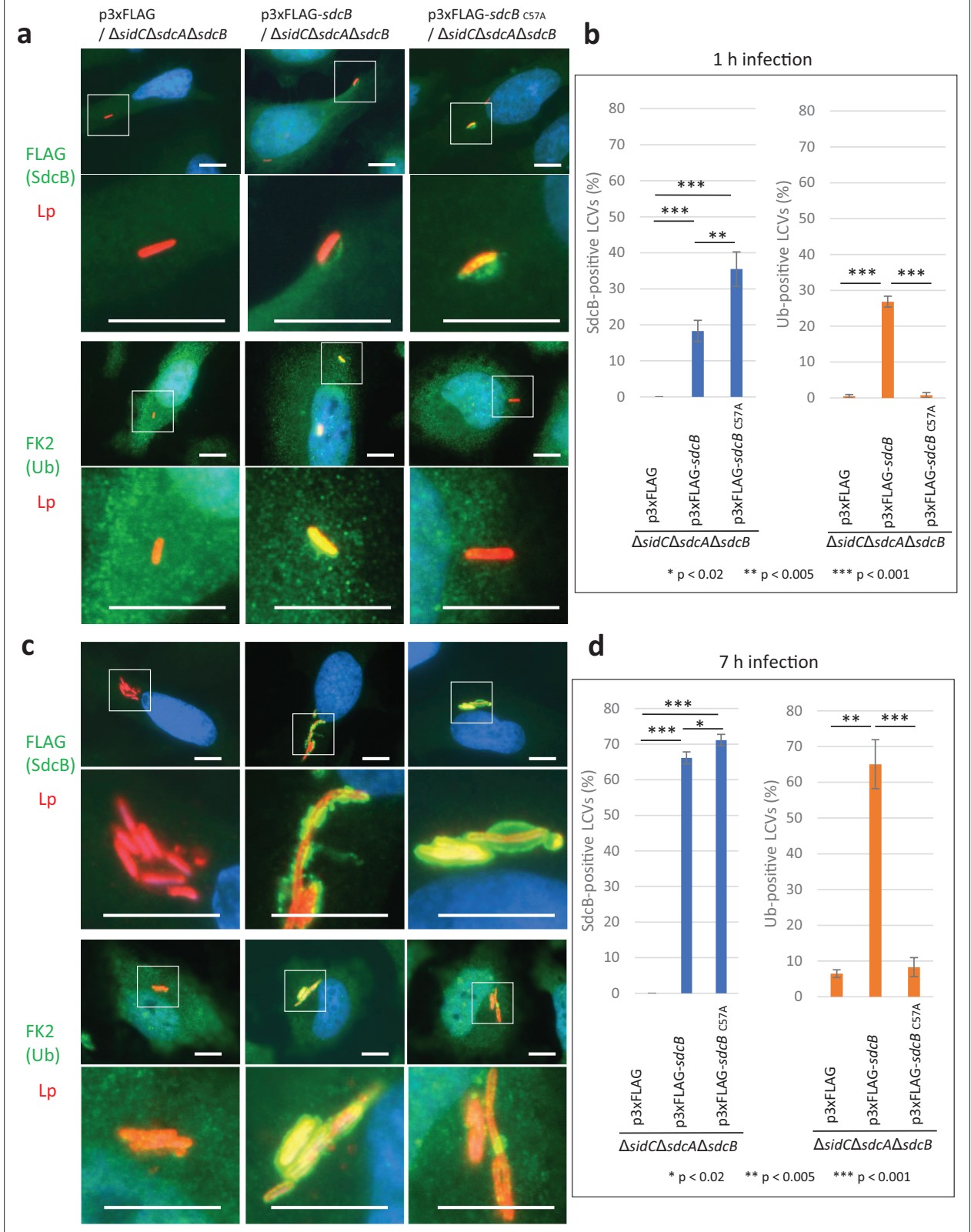

**Figure 3.** SdcB associates with the *Legionella*-containing vacuole (LCV) and plays a major role in Ub recruitment to the LCV at late stages of infection. HeLa-FcγRII cells were infected with the indicated *L. pneumophila* strains at an MOI of 2 for 1 hr (**a, b**) and for 7 hr (**c, d**). (**a, c**) Representative images of infected cells. Fixed cells were stained for FLAG-SdcB or Ub (green), *L. pneumophila* (red), and DNA (blue). Magnified images in the white squares are

*Figure 3 continued on next page*

*Figure 3 continued*

shown in the lower panels. Scale bars, 10 μm. (**b, d**) Quantitation of SdcB-positive (left) and of Ub-positive (right) LCVs (%). Infections were performed in triplicate and each value represents scoring from 200 LCVs. Significance was determined using Student's *t*-test.

The online version of this article includes the following source data for figure 3:

**Source data 1.** Raw images of micrographs in *Figure 3a*.

**Source data 2.** Raw images of micrographs in *Figure 3c*.

**Source data 3.** Counting data in *Figure 3, d*.

SdcB bands, suggesting that auto-ubiquitinated SdcB was not subjected to proteasomal degradation (*Figure 5—figure supplement 2a*), but the auto-ubiquitination rather caused its apparent loss by the band distribution on the gel. The catalytic activity-dependent reduction of apparent SdcB level was also detected in infected cells (*Figure 4c*, bottom panel). The transglutaminase activity of MavC is likely required for conjugation of Ub to SdcB, as the catalytic mutant of MavC (MavC $_{C74A}$) failed to modify SdcB (*Figure 5a*). We also found that unmodified SdcB was readily detected when the SdcB C57 active site was mutated regardless of the presence of MavC (*Figure 5b*, top panel), consistent with the disappearance of the band was linked to the auto-ubiquitination ability. Interestingly, the MavC-mediated Ub conjugation to SdcB$_{C57A}$ was not readily detected (*Figure 5b*). This indicates that the catalytic residue C57 of SdcB is crucial in being modified by MavC. As suggested by the reduction of the high molecular weight smears (*Figure 5a*), it is plausible that MavC suppresses the E3 ligase activity of SdcB by the unique chemical modification.

## The transglutaminase activity of MavC mediates Ub conjugation to SdcB and SdcA

To confirm the direct involvement of MavC in the unique Ub modification of SdcB, we reconstructed an in vitro reaction using purified proteins. In the presence of MavC, but not of its paralog MvcA, the mass shift of SdcB was readily detected (*Figure 5c*, top panel). The immunoblotting showed that the band contains Ub (*Figure 5C*, middle panel). The presence of wild-type MavC, but not its catalytic mutant, also reduced SdcB auto-ubiquitination in vitro (*Figure 5—figure supplement 2b*). Since SidC, SdcA, and SdcB are paralogs to each other, we examined if SidC and/or SdcA are also subject to MavC-mediated Ub conjugation. Triple-FLAG-tagged SidC or SdcA were transiently expressed together with GFP-MavC and HA-Ub in HEK293T-FcγRII cells, and the 3xFLAG-tagged proteins were immunoprecipitated (*Figure 5d*). These results clearly demonstrated that Ub was conjugated to SdcA but not to SidC by the catalytic activity of MavC, and that this modification occurred only to SdcA in the catalytically active form.

We then analyzed the MavC-mediated Ub conjugation to SdcB using derivatives of Ub (*Figure 5e, f*). The use of Ub having no Lys residues (HA-Ub $_{No K}$) resulted in an enhanced level of Ub conjugation to SdcB mediated by functional MavC (*Figure 5e*, middle panel). Surprisingly, HA-Ub$_{AA}$ did not conjugate to SdcB, indicating that the C-terminal Gly-Gly residues are essential for MavC-mediated Ub conjugation to SdcB (*Figure 5f*, second panel). The shifted band detected by FLAG probing represents conjugation of cellular intrinsic Ub (*Figure 5f*, top panel) as probed by Ub antibody (*Figure 5f*, third panel).

Transglutaminase activity of MavC is known to target Gln40 of Ub to catalyze the intramolecular crosslinking (*Gan et al., 2019a*; *Guan et al., 2020*; *Puvar et al., 2020*). We investigated whether the same residue of Ub is crosslinked to SdcB by the activity of MavC using mass spectrometric (MS) analysis. We found that a covalent bond was formed between Gln41 of Ub and Lys518 of SdcB (*Figure 6a*). Crosslinking between Gln31 of Ub and Lys891 of SdcB was also detected (*Figure 6—figure supplement 1*). To confirm the results, we replaced Ub residues Gln41 and Gln31 with Glu (Ub $_{Q41E}$, Ub $_{Q31E}$, and Ub $_{Q31E Q41E}$) and conducted the Ub conjugation assay by transient expression in HEK293T-FcγRII cells. Consistent with the result from the MS analysis, Ub $_{Q41E}$, but not Ub $_{Q40E}$, failed to be conjugated to SdcB, showing that Gln41 is crucial for MavC-mediated crosslinking with SdcB (*Figure 6b*, middle panel). Ub $_{Q31E}$ also reduced the level of modified SdcB, and Ub $_{Q31E Q41E}$ completely abolished the crosslinking to SdcB. The presence of modified SdcB bands when probed with anti-FLAG antibody is thought to be caused by conjugation with intrinsic Ub in cells (*Figure 6b*, top panel). Contrarily, replacement of Lys518 and Lys891 of SdcB to Arg (SdcB $_{K518R}$, SdcB $_{K891R}$, and SdcB $_{K518R K891R}$), which have no apparent effect on Rab10 ubiquitination (*Figure 6—figure supplement 2*), showed

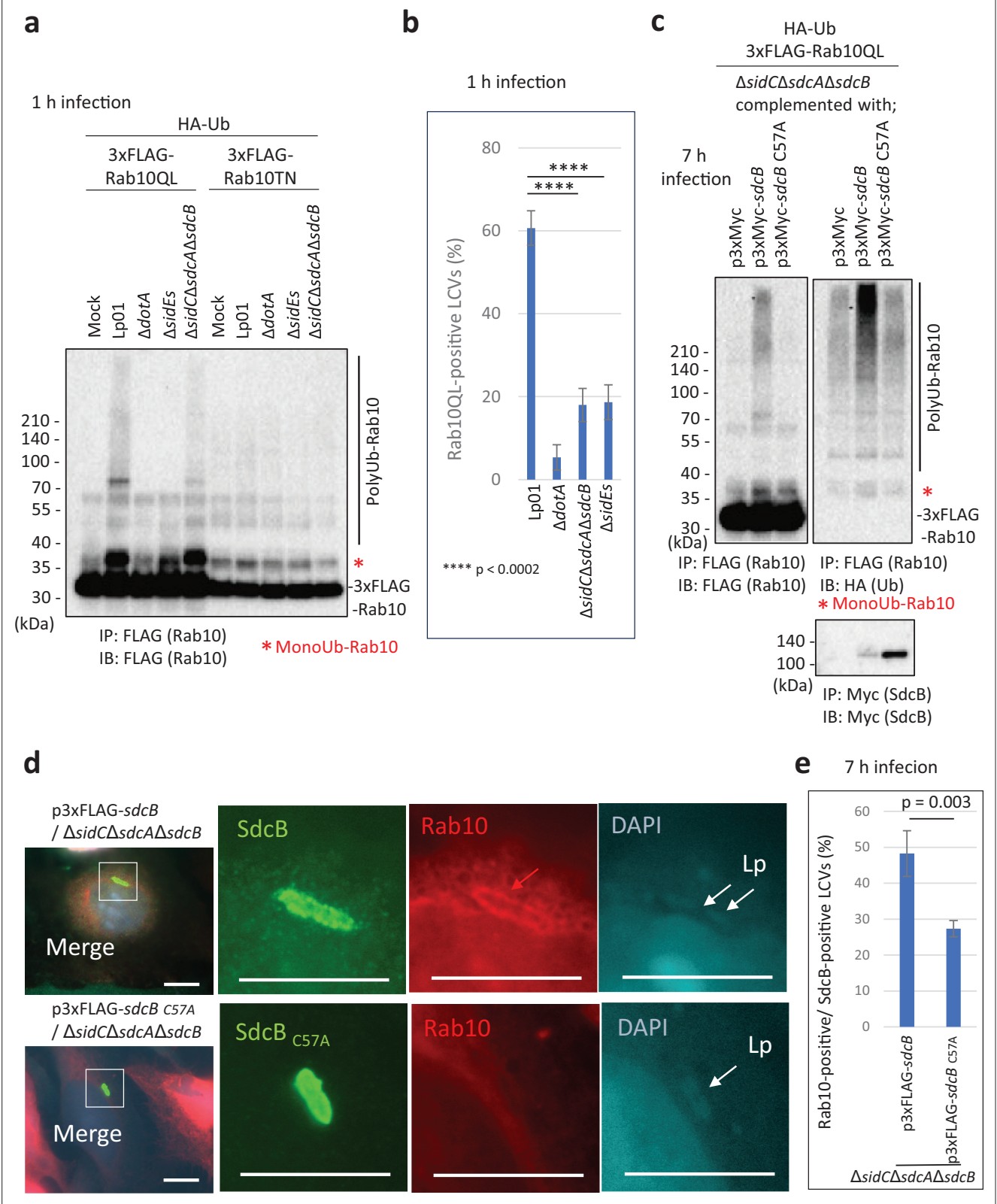

**Figure 4.** The catalytic activity of SdcB enhances retention of Rab10 on the *Legionella*-containing vacuole (LCV). (**a**) HEK293T-FcγRII cells transiently expressing 3xFLAG-Rab10 $_{Q68L}$ (QL) or Rab10 $_{T23N}$ (TN) with HA-Ub were infected with the indicated *L. pneumophila* strains for 1 hr at an MOI of 50. Rab10 was isolated from cell lysate by immunoprecipitation using anti-FLAG magnetic beads and was probed with anti-FLAG antibody. (**b**) HeLa-FcγRII cells transiently expressing RFP-Rab10QL were infected with the indicated *L. pneumophila* strains at an MOI of 10 for 1 hr (see *Figure 4—figure*

*Figure 4 continued on next page*

*Figure 4 continued*

supplement 2). Rab10-positive LCVs (%) were quantified. Infections were performed in triplicate and each value represents scoring from 50 LCVs. Significance was determined using Student's *t*-test and represented as: ****p < 0.0002. (**c**) HEK293T-FcγRII cells transiently expressing 3xFLAG-Rab10QL and HA-Ub were infected with the *L. pneumophila* strains expressing Myc-tagged SdcB or its catalytic mutant for 7 hr at an MOI of 20. Rab10 was isolated from cell lysate by immunoprecipitation using anti-FLAG magnetic beads and was probed with anti-FLAG and with anti-HA antibodies. For detection of translocated SdcB, it was isolated from cell lysate by immunoprecipitation using anti-Myc magnetic beads and was probed with anti-Myc antibody. Note that apparent reduction of the wild-type SdcB was caused by its auto-ubiquitination leading to the molecular weight shift (see text). (**d, e**) HeLa-FcγRII cells transiently expressing RFP-Rab10 were infected with the indicated *L. pneumophila* strains at an MOI of 2 for 7 hr. (**d**) Representative images of infected cells. Fixed cells were stained for FLAG-SdcB (green) and *L. pneumophila* (blue) and visualized with RFP-Rab10 (red). Magnified images in the white squares are shown in each channel. White arrows indicate the position of a bacterium. The red arrow indicates a Rab10 signal surrounding an LCV. Scale bars, 10 μm. (**e**) Quantitation of Rab10-positive LCVs (%) out of SdcB-positive ones. Infections were performed in triplicate and each value represents scoring from 200 SdcB-positive LCVs. Significance was determined using Student's *t*-test.

The online version of this article includes the following source data and figure supplement(s) for figure 4:

**Source data 1.** Original files for the western blot analysis in *Figure 4a* (anti-FLAG).

**Source data 2.** PDF containing *Figure 4a* and an original scan of the relevant western blot analysis (anti-FLAG), with cropped areas.

**Source data 3.** Original files for the western blot analysis in *Figure 4c* (anti-FLAG, anti-HA, and anti-Myc).

**Source data 4.** PDF containing *Figure 4c* and original scans of the relevant western blot analysis (anti-FLAG, anti-HA, and anti-Myc), with cropped areas.

**Source data 5.** Raw images of micrographs in *Figure 4d*.

**Source data 6.** Counting data in *Figure 4b, e*.

**Figure supplement 1.** Active Rab10 is preferentially targeted for infection-induced ubiquitination.

**Figure supplement 1—source data 1.** Original files for the western blot analysis in *Figure 4—figure supplement 1* (anti-FLAG and anti-HA).

**Figure supplement 1—source data 2.** PDF containing *Figure 4—figure supplement 1* and original scans of the relevant western blot analysis (anti-FLAG and anti-HA), with cropped areas.

**Figure supplement 2.** The SidE- and SidC-family proteins contribute toward retaining active Rab10 to the *Legionella*-containing vacuole (LCV).

**Figure supplement 2—source data 1.** Raw images of in micrographs in *Figure 4—figure supplement 2*.

lesser impact on abolishing the reactivity (*Figure 6c*), suggesting that additional residues of SdcB can be subjected to MavC-dependent Ub conjugation. Taken together, we currently hypothesize that association of the C-terminal Gly of Ub to the catalytic pocket of SdcA or SdcB (C44 or C57, respectively) positions these molecules in proper orientation for intramolecular crosslinking mediated by the transglutaminase activity of MavC.

## Catalytic activity of MavC can impact Rab10 localization to the LCV

We then examined the role of MavC in the LCV localization of Rab10. By immunofluorescent microscopy, we monitored the level of RFP-tagged Rab10 on SdcB-positive LCVs when MavC or its catalytic mutant were expressed in HeLa-FcγRII cells (*Figure 7a*). At 4 hr after infection with *L. pneumophila* strains expressing 3xFLAG-SdcB, the level of Rab10-positive LCVs was significantly higher in the cells expressing the catalytic mutant of MavC compared with those expressing wild-type MavC (*Figure 7a, b*). That Rab10 localization was reduced coincident with MavC-dependent inhibition of SdcB further supports the contribution of SdcB activity toward retention of Rab10 on the LCV. We then wondered whether bacterially delivered MavC can contribute to the elimination of Rab10 from the vacuole. As the levels of the Rab10-positive LCVs were not significantly altered up to 7 hr after infection with the wild-type *L. pneumophila* strain (*Figure 2b*), we examined Rab10 localization at a later time point after infection (*Figure 7c*, *Figure 7—figure supplement 1*). At 9 hr after infection with the wild-type strain, the level of Rab10-positive LCVs was reduced to about 20% of the total LCVs (compare with that of 1–7 hr infection (~40%)) (*Figure 2b*). However, the level got significantly higher in the cells infected with a Δ*mavC*Δ*mvcA* strain. Contrary, when the cells were infected with a strain lacking Lpg2149 which inhibits the activity of MavC and MvcA, the level declined. These results support that MavC suppresses the activity of SdcB in infection conditions and thereby downregulates Rab10 localization to the LCV at later stages of infection. Taken together, we found that Rab10 localization is finely regulated during infection; the interplay of bacterial enzymes leads to sustained association of Rab10 with the LCV and eventually dissociates it from the LCV presumably in accordance with the process of the LCV biogenesis (*Figure 7d*).

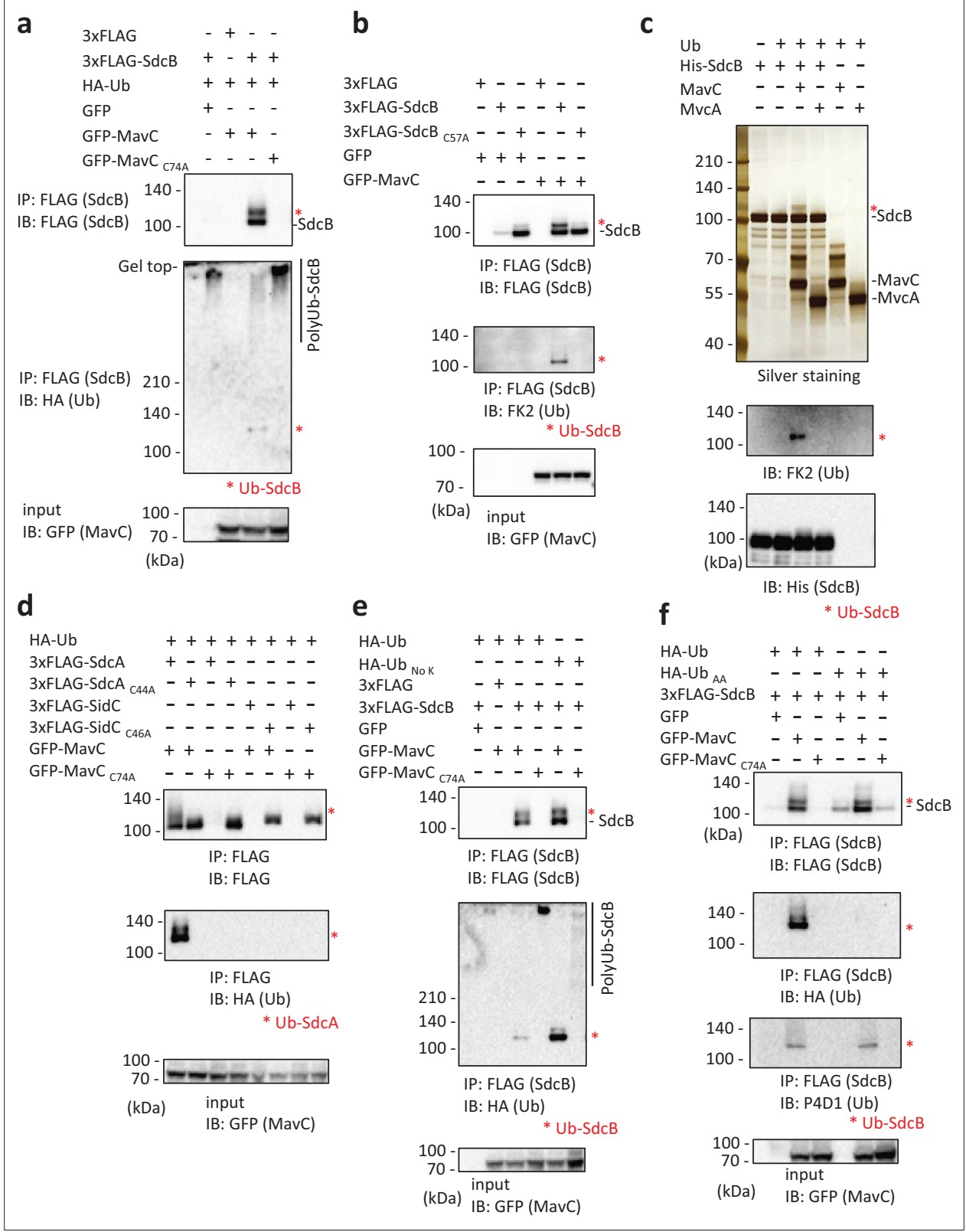

**Figure 5.** The transglutaminase activity of MavC can mediate a unique Ub conjugation to SdcB. (**a**) 3xFLAG-SdcB, HA-Ub, and GFP-MavC were coexpressed in HEK293T-FcγRII cells. SdcB was isolated from cell lysates by immunoprecipitation using anti-FLAG magnetic beads and was probed with the indicated antibodies. The asterisks indicate the Ub-conjugated form of SdcB. (**b**) 3xFLAG-SdcB and GFP-MavC were coexpressed in HEK293T-FcγRII cells. SdcB was isolated from cell lysates by immunoprecipitation using anti-FLAG magnetic beads and was probed with the indicated antibodies.

*Figure 5 continued on next page*

*Figure 5 continued*

The asterisks indicate the Ub-conjugated form of SdcB. (**c**) In vitro transglutaminase assay was performed using purified proteins. The samples were analyzed by sodium dodecyl sulfate–polyacrylamide gel electrophoresis (SDS–PAGE) followed by silver staining (top) or by immunoblotting using the indicated antibodies (middle and bottom). The asterisks indicate the Ub-conjugated form of SdcB. (**d**) 3xFLAG-SdcA or SidC, GFP-MavC, and HA-Ub were coexpressed in HEK293T-FcγRII cells. SdcA or SidC was isolated from cell lysates by immunoprecipitation using anti-FLAG magnetic beads and was probed with the indicated antibodies. The asterisks indicate the Ub-conjugated form of SdcA. (**e**) 3xFLAG-SdcB, GFP-MavC, and HA-Ub or Ub without any Lys residues (Ub $_{No K}$) were coexpressed in HEK293T-FcγRII cells. SdcB was isolated from cell lysates by immunoprecipitation using anti-FLAG magnetic beads and was probed with the indicated antibodies. The asterisks indicate the Ub-conjugated form of SdcB. (**f**) 3xFLAG-SdcB, GFP-MavC, and HA-Ub or Ub in which the C-terminal GG were replaced with AA (Ub $_{AA}$) were coexpressed in HEK293T-FcγRII cells. SdcB was isolated from cell lysates by immunoprecipitation using anti-FLAG magnetic beads and was probed with the indicated antibodies. The asterisks indicate the Ub-conjugated form of SdcB.

The online version of this article includes the following source data and figure supplement(s) for figure 5:

**Source data 1.** Original files for the western blot analysis in *Figure 5a* (anti-FLAG, anti-HA, and anti-GFP).

**Source data 2.** PDF containing *Figure 5a* and original scans of the relevant western blot analysis (anti-FLAG, anti-HA, and anti-GFP), with cropped areas.

**Source data 3.** Original files for the western blot analysis in *Figure 5b* (anti-FLAG, anti-FK2, and anti-GFP).

**Source data 4.** PDF containing *Figure 5b* and original scans of the relevant western blot analysis (anti-FLAG, anti-FK2, and anti-GFP), with cropped areas.

**Source data 5.** Original files for the silver stained gel and western blot analysis in *Figure 5c* (anti-FK2 and anti-His).

**Source data 6.** PDF containing *Figure 5c* and original scans of the relevant silver stained gel and western blot analysis (anti-FK2 and anti-His), with cropped areas.

**Source data 7.** Original files for the western blot analysis in *Figure 5d* (anti-FLAG, anti-HA, and anti-GFP).

**Source data 8.** PDF containing *Figure 5d* and original scans of the relevant western blot analysis (anti-FLAG, anti-HA, and anti-GFP), with cropped areas.

**Source data 9.** Original files for the western blot analysis in *Figure 5e* (anti-FLAG, anti-HA, and anti-GFP).

**Source data 10.** PDF containing *Figure 5e* and original scans of the relevant western blot analysis (anti-FLAG, anti-HA, and anti-GFP), with cropped areas.

**Source data 11.** Original files for the western blot analysis in *Figure 5f* (anti-FLAG, anti-HA, anti-P4D1, and anti-GFP).

**Source data 12.** PDF containing *Figure 5f* and original scans of the relevant western blot analysis (anti-FLAG, anti-HA, anti-P4D1, and anti-GFP), with cropped areas.

**Figure supplement 1.** SdcB has a catalytic activity of self-ubiquitination with preference of various E2 enzymes.

**Figure supplement 1—source data 1.** Original files for the silver staining and the western blot analysis in *Figure 5—figure supplement 1* (silver staining and anti-FK2).

**Figure supplement 1—source data 2.** PDF containing original scans of the silver stained gel and the western blot analysis in *Figure 5—figure supplement 1* (anti-FK2) with cropped areas.

**Figure supplement 2.** The catalytic activity of MavC negatively impacts on auto-ubiquitination of SdcB.

**Figure supplement 2—source data 1.** Original files for the western blot analysis in *Figure 5—figure supplement 2a* (anti-FLAG, anti-GFP and anti-GAPDH).

**Figure supplement 2—source data 2.** PDF containing *Figure 5—figure supplement 2a* and original scans of the relevant western blot analysis (anti-FLAG, anti-GFP, and anti-GAPDH), with cropped areas.

**Figure supplement 2—source data 3.** Original files for the western blot analysis in *Figure 5—figure supplement 2b* (anti-FK2).

**Figure supplement 2—source data 4.** PDF containing *Figure 5—figure supplement 2b* and original scans of the relevant western blot analysis (anti-FK2), with cropped areas.

## Discussion

Following the finding that subversion of Rab1 function is critical for *L. pneumophila* to create the replicative vacuole (*Kagan et al., 2004*), remarkable numbers of studies have revealed the molecular mechanisms of how *L. pneumophila* T4SS effectors modulate the localization and enzymatic activity of Rab1 (*Qiu and Luo, 2017*). The importance of Rab10 and its chaperone *RABIF* for intracellular replication of *L. pneumophila* has recently emerged (*Jeng et al., 2019*). However, little is known about how *L. pneumophila* manipulates the activity of Rab10.

We found that the SidE-family effectors mediate noncanonical ubiquitination of Rab10. Apparently, this modification precedes and is prerequisite for subsequent polyubiquitination of Rab10 (*Figure 1*),

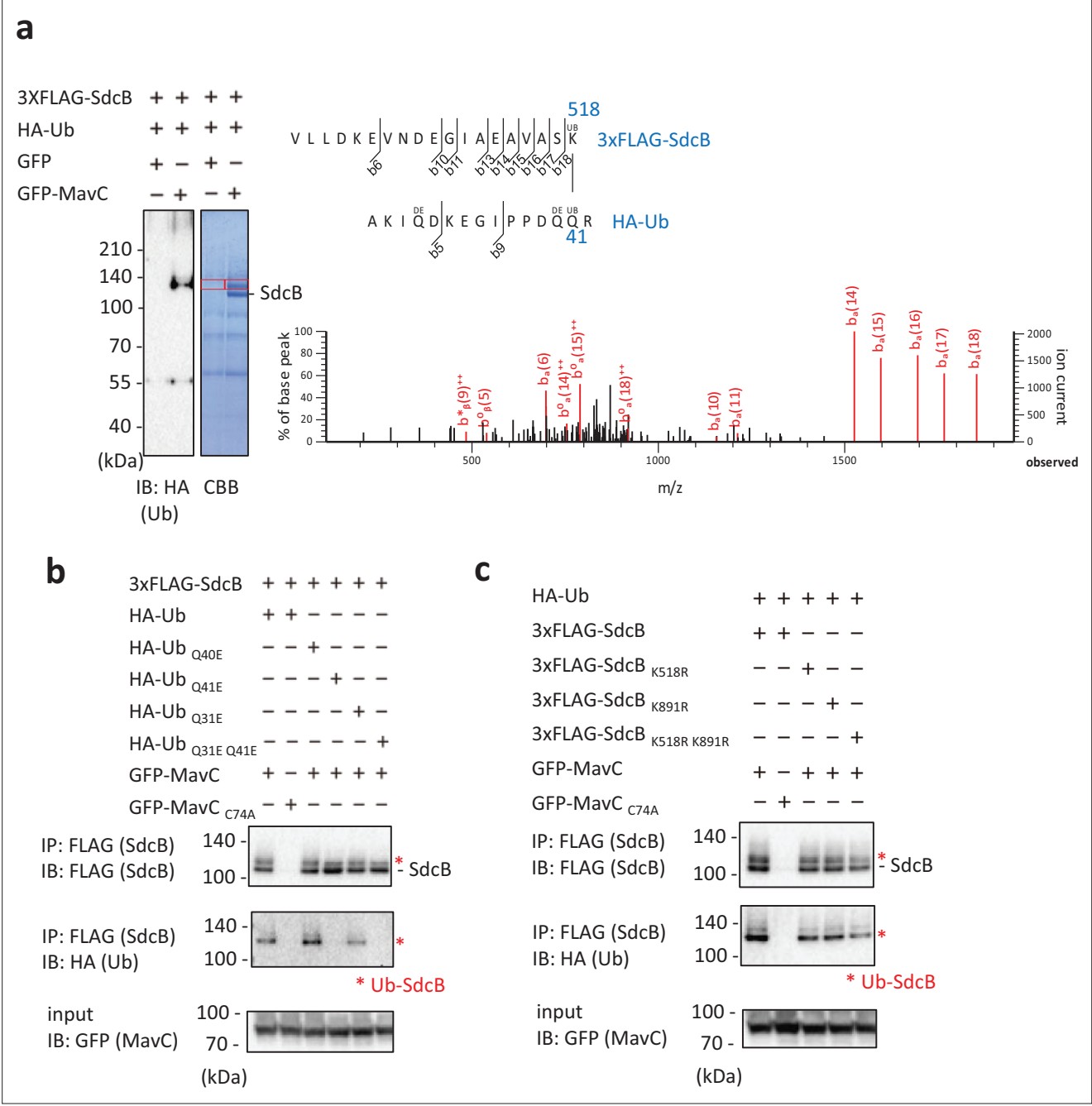

**Figure 6.** Identification of residues on Ub and SdcB between which MavC can crosslink. (**a**) MavC catalyzes the formation of an isopeptide bond between the Gln41 of Ub and the Lys518 of SdcB. The indicated proteins were expressed in HEK293T-FcγRII cells and SdcB was isolated from cell lysates by immunoprecipitation using anti-FLAG magnetic beads. The samples were resolved by sodium dodecyl sulfate–polyacrylamide gel electrophoresis (SDS–PAGE). The Ub-conjugated SdcB was detected by immunoblotting and by CBB staining. The gel slices of areas of the bands shown with the red squares were subjected to mass spectrometric analysis. Product ion spectrum was shown for Ub peptide – AKIQDKEGIPPDQQR crosslinked with SdcB peptide – VLLDKEVNDEGIAEAVASK. (**b, c**) The indicated proteins were coexpressed in HEK293T-FcγRII cells. SdcB was isolated from cell lysates by immunoprecipitation using anti-FLAG magnetic beads and was probed with the indicated antibodies. The asterisks indicate the Ub-conjugated form of SdcB.

The online version of this article includes the following source data and figure supplement(s) for figure 6:

**Source data 1.** Original files for the CBB stained gel and western blot analysis in *Figure 6a* (anti-HA).

**Source data 2.** PDF containing *Figure 6a* and original scans of the relevant CBB stained gel and western blot analysis (anti-HA), with cropped areas.

**Source data 3.** Original files for the western blot analysis in *Figure 6b* (anti-FLAG, anti-HA, and anti-GFP).

*Figure 6 continued on next page*

*Figure 6 continued*

**Source data 4.** PDF containing *Figure 6b* and original scans of the relevant western blot analysis (anti-FLAG, anti-HA, and anti-GFP), with cropped areas.

**Source data 5.** Original files for the western blot analysis in *Figure 6c* (anti-FLAG, anti-HA, and anti-GFP).

**Source data 6.** PDF containing *Figure 6c* and original scans of the relevant western blot analysis (anti-FLAG, anti-HA, and anti-GFP), with cropped areas.

**Figure supplement 1.** Mass spectrometry analysis identified additional residues forming a covalent linkage between Ub and SdcB.

**Figure supplement 2.** Mutations on Lys518 and Lys891 of SdcB did not affect ubiquitination of Rab10.

**Figure supplement 2—source data 1.** Original files for the western blot analysis in *Figure 6—figure supplement 2* (anti-FLAG, anti-HA, and anti-Myc).

**Figure supplement 2—source data 2.** PDF containing *Figure 6—figure supplement 2* and original scans of the relevant western blot analysis (anti-FLAG, anti-HA, and anti-Myc), with cropped areas.

which is linked to its localization to the LCV (*Figure 2*). Polyubiquitination of Rab10 was enhanced in a manner depending on the catalytic activities of the SidC-family proteins (*Figures 1 and 4a, c*) consistent with previous reports (*Jeng et al., 2019*; *Liu et al., 2020*). As ectopic expression of SdeA in HEK293T-FcγRII cells led to monoubiquitination of Rab10 (*Figure 7—figure supplement 2*), it is plausible that PR-Ub conjugation to Rab10 is directly catalyzed by the SidE-family ligases. However, the SdcB-mediated polyubiquitination of Rab10 was not readily detected even in the presence of SdeA lacking its DUB domain (*Figure 7—figure supplement 2*). This result prompted us to consider that polyubiquitination of Rab10 by the SidC-family ligases can occur only in a specific circumstance, i.e. on the LCV. It is also possible that Rab10 can be partly catalyzed by canonical E3 ligases in host cells and/or other *L. pneumophila* effector proteins. The Ub ligase activity of SdcB strongly induced accumulation of Ub on the LCV (*Figure 3*), showing that Rab10 may not be an only substrate of SdcB. The PR-Ub modification of Rab10 and ubiquitination of unknown LCV-associated substrates of the SidC family may provide the platform for the further modification of Rab10.

Involvement of MavC for regulation of the SidC-family ligases was unexpectedly identified in our analyses. This finding has added another layer of complexity to effector-mediated regulation of Rab10. It was previously demonstrated that MavC can catalyze covalent linkage between Q40 of Ub with a Lys residue of UBE2N (*Gan et al., 2019a*). Our result showing that Q41 of Ub is crosslinked with SdcB gives a new insight into the molecular mechanism of how MavC catalyzes Ub conjugation to specific substrates. In addition, both the C-terminal Gly of Ub (*Figure 5f*) and the catalytic Cys of SdcB (C57A) (*Figure 5b*) were essential for crosslinking. This strongly suggests that the ability of SdcB as a Ub ligase to position a Ub molecule into its active site is a requirement for MavC to form the covalent bond between Ub and SdcB. This model is reminiscent of the reaction scheme described for how MavC mediates Ub conjugation to UBE2N, in which the initial capturing of Ub by the E2 activity of UBE2N allows enhanced activity by MavC (*Puvar et al., 2020*). We investigated SdcB in terms of the MavC-mediated regulation in this study and found that suppression of its activity can be beneficial for release of Rab10 at the maturation stage of LCV biogenesis (*Figure 7d*). However, MavC may have multiple targets including SdcA (*Figure 5d*) which can be involved in the early stage of infection. How MavC regulates the entire process of the vacuolar modification needs to be addressed in future studies.

In contrast to the PI[4]P-binding domain present in SidC and SdcA, SdcB has an ankyrin repeat (ANK) domain at its C-terminus. As ANK domains are generally known to mediate protein–protein interactions, we speculate that SdcB targets substrates distinct from those of SidC and SdcA. The lipid-binding ability of SidC and SdcA implicates their substrate specificity; their substrates are on or associated with the LCV in the specific stages of vacuole remodeling (*Vormittag et al., 2023*). We, therefore, speculate that SdcB can target substrate(s) present on the LCV in maturation stages different from ones when SidC and SdcA can work in. In spite of our extensive efforts, we have not succeeded to identify cellular targets with which the ANK domain of SdcB interacts. It would be a future perspective to understand the exact biological role of SdcB as a Ub ligase in *L. pneumophila* infection by identifying its enzymatic substrates.

We found that *L. pneumophila* has a multi-tiered regulatory mechanism to manipulate Ub signaling cascades in which SidC- and SidE-family Ub ligases are involved, even though the exact molecular

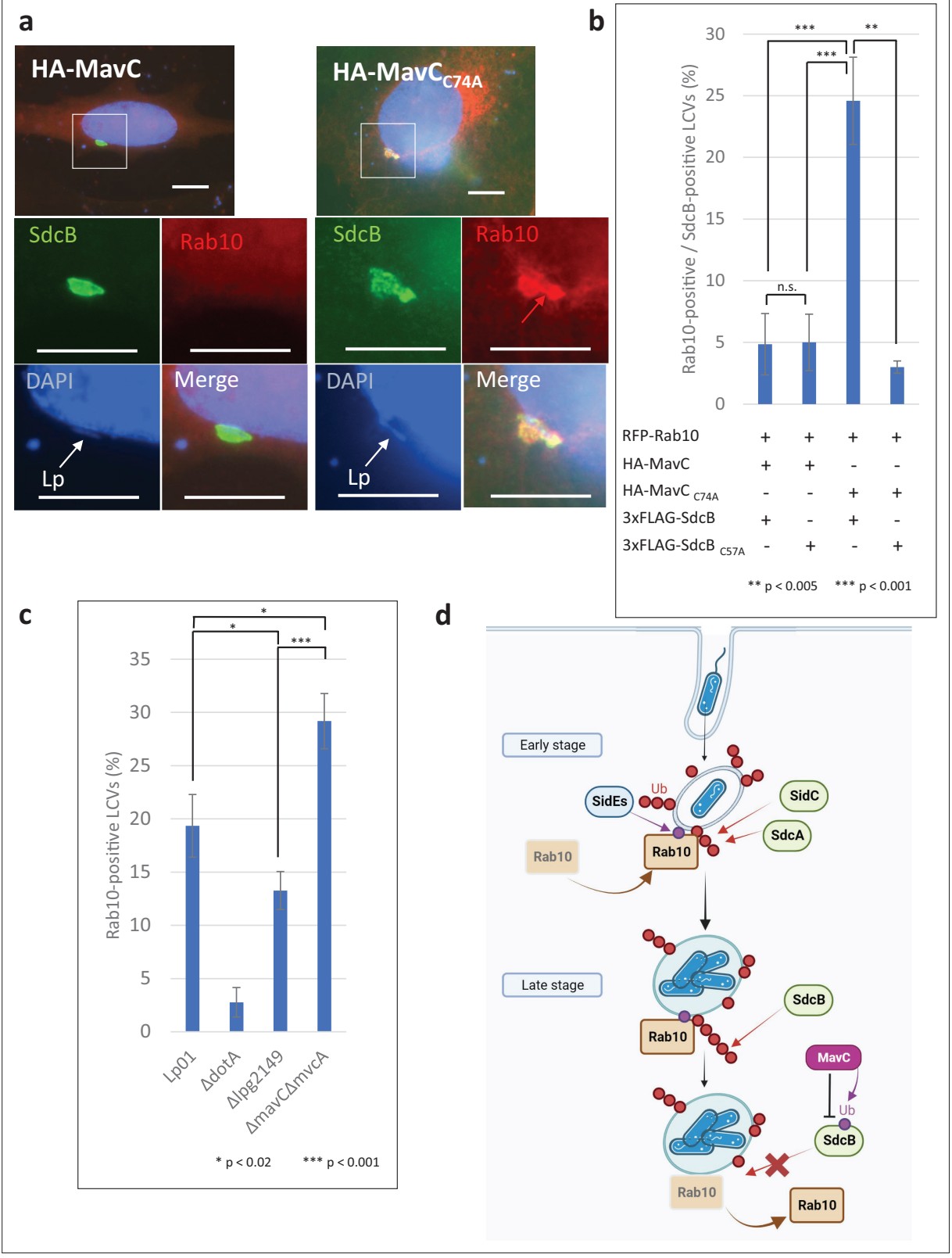

**Figure 7.** Catalytic activity of MavC negatively regulates the Rab10 localization to the *Legionella*-containing vacuole (LCV). (**a, b**) HeLa-FcγRII cells transiently expressing RFP-Rab10 and HA-MavC or its catalytic mutant were infected with the Lp01 Δ*sidC*Δ*sdcA*Δ*sdcB* strain complemented with the plasmid expressing 3xFLAG-SdcB or its catalytic mutant at an MOI of 2 for 4 hr. (**a**) Representative images of cells infected with the Lp01 Δ*sidC*Δ*sdcA*Δ*sdcB* strain complemented with the plasmid expressing 3xFLAG-SdcB. Fixed cells were stained for FLAG-SdcB (green) and DNA (blue),

*Figure 7 continued on next page*

*Figure 7 continued*

and visualized with RFP-Rab10 (red). Magnified images in the white squares are shown in each channel. White arrows indicate the position of a bacterium. The red arrow indicates a Rab10 signal surrounding an LCV. Scale bars, 10 μm. (**b**) Quantitation of Rab10-positive LCVs (%) out of SdcB-positive ones. Infections were performed in triplicate and each value represents scoring from 200 SdcB-positive LCVs. Significance was determined using Student's *t*-test. (**c**) HeLa-FcγRII cells transiently expressing RFP-Rab10 were infected with the indicated Lp01 strains at an MOI of 2 for 9 hr, and Rab10-positive LCVs (%) were quantified. Infections were performed in triplicate and each value represents scoring from 200 LCVs. Significance was determined using Student's *t*-test. (**d**) The schematic of roles of SidE- and SidC-family ligases in Rab10 localization to the LCV and of negative regulation of SdcB-dependent Rab10 retention by the transglutaminase activity of MavC. Red arrows indicate canonical Ub conjugation by SidC, SdcA, and SdcB. Purple arrows indicate the noncanonical Ub conjugation. In the early stage of infection, Rab10 is recruited and retained to the LCV. This event is linked to its phosphoribosylated (PR) ubiquitination catalyzed by the SidE effectors. The PR ubiquitination of Rab10 provides a platform of its polyubiquitination in a manner depending on SidC and SdcA. In later stages, SdcB contributes toward sustained Ub accumulation on the LCV, enabling the LCV to maintain Rab10 on the vacuole. MavC-mediated crosslinking between Ub and SdcB disrupts the catalytic activity of SdcB, eventually releasing Rab10 from the LCV.

The online version of this article includes the following source data and figure supplement(s) for figure 7:

**Source data 1.** Raw images of micrographs in *Figure 7a*.

**Source data 2.** Counting data in *Figure 7b, c*.

**Figure supplement 1.** Bacterially delivered MavC mediates elimination of Rab10 from the *Legionella*-containing vacuole (LCV).

**Figure supplement 1—source data 1.** Raw images of in micrographs in *Figure 7—figure supplement 1*.

**Figure supplement 2.** Ectopic expression of SdcB does not proceed Rab10 polyubiquitination even in the presence of SdeA.

**Figure supplement 2—source data 1.** Original files for the western blot analysis in *Figure 7—figure supplement 2* (anti-GFP, anti-RFP, and anti-HA).

**Figure supplement 2—source data 2.** PDF containing *Figure 7—figure supplement 2* and original scans of the relevant western blot analysis (anti-GFP, anti-RFP, and anti-HA), with cropped areas.

mechanism how Ub modification of Rab10 results in regulation of its LCV residency would be an open question. MavC was found to contribute to fine-tuning the regulation via its unique Ub conjugation activity toward SdcB. These regulations are reflected in many aspects of the LCV biogenesis and maturation which are thought to be largely controlled by Ub signaling. The regulatory cascade of Rab10 GTPase, whose function is crucial for ER recruitment to the LCV, became apparent in this study. Most likely, similar regulatory cascades exist for many LCV-associated proteins.

# Materials and methods

**Key resources table**

| Reagent type (species) or resource | Designation | Source or reference | Identifiers | Additional information |
|---|---|---|---|---|
| Strain, strain background (*Legionella pneumophila*) | Philadelphia-1 (Lp01) | *Berger and Isberg, 1993* | NC_002942.5 | |
| Strain, strain background (*Legionella pneumophila*) | Lp01ΔicmV ΔdotA (ΔdotA) | *Zuckman et al., 1999* | N/A | |
| Strain, strain background (*Legionella pneumophila*) | Lp01 ΔsidCΔsdcA | This study | N/A | Constructed in Nagai lab |
| Strain, strain background (*Legionella pneumophila*) | Lp01 ΔsidCΔsdcAΔsdcB | This study | N/A | Constructed in Nagai lab |
| Strain, strain background (*Legionella pneumophila*) | Lp01 ΔsidEΔsdeAΔsdeBΔsdeC (ΔsidEs) | This study | N/A | Constructed in Nagai lab |
| Strain, strain background (*Legionella pneumophila*) | Lp01 ΔdupAΔdupB | This study | N/A | Constructed in Nagai lab |
| Strain, strain background (*Legionella pneumophila*) | Lp01 ΔdupAΔsidJΔdupBΔsdjA | This study | N/A | Constructed in Nagai lab |
| Strain, strain background (*Legionella pneumophila*) | Lp01 Δlpg2149 | This study | N/A | Constructed in Nagai lab |
| Strain, strain background (*Legionella pneumophila*) | Lp01 ΔmavCΔmvcA | This study | N/A | Constructed in Nagai lab |

*Continued on next page*

*Continued*

| Reagent type (species) or resource | Designation | Source or reference | Identifiers | Additional information |
|---|---|---|---|---|
| Strain, strain background (*Escherichia coli*) | DH5α | TOYOBO | Cat# DNA-903 | Competent cells |
| Strain, strain background (*Escherichia coli*) | DH5α *λ pir* | *Zuckman et al., 1999* | N/A | Competent cells |
| Strain, strain background (*Escherichia coli*) | BL21(DE3) | NOVAGEN-MERK | Cat# 69450 | Competent cells |
| Cell line (*Homo sapiens*) | HeLa-FcγRII | *Arasaki et al., 2017* | Established from ATCC CCL-2 | |
| Cell line (*Homo sapiens*) | HEK293T-FcγRII | *Arasaki and Roy, 2010* | Established from ATCC CRL-3216 | |
| Antibody | anti-FLAG (M2) (Mouse monoclonal) | Sigma | Cat# F1804 | WB (1:1000) |
| Antibody | anti-HA (Mouse monoclonal) | MBL | Cat# M132-3 | WB (1:1000) |
| Antibody | anti-HA (Rabbit monoclonal) | MBL | Cat# 561 | WB (1:1000) |
| Antibody | anti-Ub (FK2) (Mouse monoclonal) | Enzo | Cat# BML-PW8810 | WB (1:1000) |
| Antibody | anti-Ub (P4D1) (Mouse monoclonal) | Santa Cruz | Cat# sc-8017 | WB (1:200) |
| Antibody | anti-GFP (Rabbit polyclonal) | MBL | Cat# 598 | WB (1:2000) |
| Antibody | anti-His (Mouse monoclonal) | Novagen | Cat# 70796-3 | WB (1:1000) |
| Antibody | anti-Myc (Mouse monoclonal) | Roche | Cat# 11 667 203 001 | WB (1:1000) |
| Antibody | anti-RFP (Rabbit polyclonal) | MBL | Cat# PM005 | WB (1:1000) |
| Antibody | Anti-GAPDH (Mouse monoclonal) | Proteintec | Cat# 60004-1-Ig | WB (1:5000) |
| Antibody | anti-*Legionella pneumophila* (Rabbit polyclonal) | BioAcademia | Cat# 64-100 | IF (1:5000) Opsonization (1:3000) |
| Antibody | Goat anti-mouse IgG (H+L) secondary, HRP | Thermo Fisher | Cat# 62-6520 | WB (1:10,000) |
| Antibody | Goat anti-rabbit IgG (H+L) secondary, HRP | Thermo Fisher | Cat# 65-6120 | WB (1:10,000) |
| Antibody | Alexa Fluor 488 goat anti-mouse | Thermo Fisher | Cat#A-11029 | IF (1:500) |
| Antibody | Alexa Fluor 488 goat anti-rabbit | Thermo Fisher | Cat#A-11034 | IF (1:500) |
| Antibody | Rhodamine RedX goat anti-rabbit | Thermo Fisher | Cat# R6349 | IF (1:1000) |
| Peptide, recombinant protein | Ubiquitin, human recombinant | Boston Biochem | Cat# U-100H | |
| Peptide, recombinant protein | Ubiquitin K63R, human recombinant | Boston Biochem | Cat# UM-K63R | |
| Peptide, recombinant protein | Ubiquitin mutant with K63 only, human recombinant | Boston Biochem | Cat# UM-K630 | |
| Peptide, recombinant protein | UBE1, human recombinant | Boston Biochem | Cat# E-305 | |
| Peptide, recombinant protein | Ubc (E2) Enzyme Kit | Boston Biochem | Cat# K-980B | |
| Chemical compound, drug | *N*-(2-Acetamido)-2-aminoethanesulfonic acid (ACES) | Sigma | Cat# 7365-82-4 | |
| Chemical compound, drug | *N*-Ethylmaleimide (NEM) | Sigma | Cat# E3876 | |
| Chemical compound, drug | cOmplete protease inhibitor Cocktail (EDTA free) | Roche (Merk) | Cat# 11873580001 | |
| Chemical compound, drug | SigmaFast Protease Inhibitor Cocktail | Sigma | Cat# S8830 | |
| Chemical compound, drug | Phenylmethylsulfonyl fluoride (PMSF) | Nacarai | Cat# 27327-94 | |
| Chemical compound, drug | MG132 | Calbiochem | Cat# 474791 | |

*Continued on next page*

*Continued*

| Reagent type (species) or resource | Designation | Source or reference | Identifiers | Additional information |
|---|---|---|---|---|
| Commercial assay kit | Silver Stain MS Kit | FUJIFILM Wako | Cat# 299-58901 | |
| Commercial assay kit | QuickChange II site-directed mutagenesis kit | Agilent | Cat# 200523 | |
| Commercial assay kit | Gibson assembly kit | New England Biolabs | Cat# E2611 | |
| Commercial assay kit | EndoFree Plasmid MAXI prep kits | QIAGEN | Cat# 12362 | |
| Other | 4,6-Diamidino-2-phenylindole (DAPI) | DOJINDO | Cat# GW094 | 1:10,000 |
| Other | Lipofectamine 2000 | Invitrogen | Cat# 11668-019 | Transfection reagent |
| Other | Polyethylenimine (PEI) | Polysciences | Cat# 24765-2 | Transfection reagent |
| Other | Poly-L-lysine | Sigma | Cat# P4707 | 0.01% |
| Other | Paraformaldehyde (PFA) | Sigma | Cat# 441244 | 4% |
| Other | ProLongTM Diamond Antifade Mountant | Thermo Fisher | Cat# P36961 | Antifade moutant |
| Other | Ni-nitrilotriacetic acid (NTA) agarose | QIAGEN | Cat# 30210 | Affinity matrix |
| Other | FLAG M2 magnetic beads | Sigma | Cat# M8823 | Affinity beads |
| Other | Myc-Trap magnetic beads | chromotek | Cat# ytma | Affinity beads |
| Other | RFP-Trap magnetic beads | chromotek | Cat# rtma | Affinity beads |
| Other | Minimum essential medium α (MEMα) | Gibco | Cat# 12571-063 | Medium |
| Other | Dulbecco's modified Eagle medium (DMEM) | Gibco | Cat# 11885-084 | Medium |
| Other | Fetal bovine serum (FBS) | Sigma | Cat# 172012 | Heat inactivated, 10% |
| Other | Goat serum | Gibco | Cat# 16210-064 | 2% |

## Bacterial strains and growth conditions

The *L. pneumophila* and *Escherichia coli* strains used in this study are listed in Key resource table. Deletion strains were constructed by allelic exchange, as described previously (*Zuckman et al., 1999*). The *L. pneumophila* strains were grown at 37°C in liquid *N*-(2-acetamido)-2-aminoethanesulfonic acid (Sigma)-buffered yeast extract (AYE) media or on charcoal-yeast extract (CYE) plates (*Feeley et al., 1979*) with or without appropriate antibiotics (100 µg/ml streptomycin, 10 µg/ml chloramphenicol, and 10 µg/ml kanamycin), as described previously (*Berger et al., 1994*). The *E. coli* strains (DH5α, DH5α λ pir, and BL21[DE3]) were grown at 37°C in standard media.

## Cell culture

HeLa cells stably expressing FcγRII (HeLa-FcγRII) were established (*Arasaki et al., 2017*) from HeLa cells (ATCC; CCL-2) obtained from the RIKEN Bioresource Center (RCB0007). HEK293T cells stably expressing FcγRII (HEK293T-FcγRII) were established (*Arasaki and Roy, 2010*) from HEK293T cells (ATCC; CRL-3216) supplied from Dr. Craig Roy (Yale University). HeLa-FcγRII cells were grown in minimum essential medium α (MEMα; Gibco) supplemented with 10% fetal bovine serum (FBS; Sigma). HEK293T-FcγRII cells were grown in Dulbecco's modified Eagle medium (DMEM; Gibco) supplemented with 10% FBS. All cells were incubated at 37°C under 5% $CO_2$ condition. All cell lines were regularly tested for mycoplasma contamination.

## Plasmid construction

Plasmids used in this study are listed in *Supplementary file 1*. All cloning was conducted by PCR amplification of the desired genes using primers listed in *Supplementary file 2* from genomic DNA of *L. pneumophila* or from plasmids listed in *Supplementary file 1* followed by ligation with the vectors unless otherwise noted below. Site-directed mutagenesis was carried out using a QuickChange II site-directed mutagenesis Kit (Agilent) according to the manufacture's recommendation. For construction of pET15b-His-*sdeA*ΔDUB, the entire region of pET15b-His-*sdeA* except for the region encoding 1–199

aa of SdeA was amplified with primers 2649/2712 and then the fragment was self-ligated with a Gibson assembly kit (New England Biolabs). For construction of pmGFP-sdeA$_{\Delta DUB}$, the entire region of pmGFP-sdeA except for the region encoding 1–199 aa of SdeA was amplified with primers 2714/2715 and then the fragment was self-ligated with a Gibson assembly kit. For construction of pMMB-3xMyc-sdeA, the coding region of sdeA was amplified using primers 2658/2659 from genomic DNA of Lp01, then the fragment was ligated with a linearized vector generated by PCR using 2341/2681 based on pMMB-PicmR-3xFLAG.

## Protein purification

*E. coli* cells overproducing MavC, MvcA, or SdcB with a hexa-histidine tag were collected by centrifugation and resuspended with 50 mM Tris–HCl pH 7.5, 5 mM ethylenediamine tetraacetic acid (EDTA) containing SigmaFast Protease Inhibitor Cocktail (Sigma). Cells were disrupted, centrifuged (30,000 × g, 20 min), and the soluble fraction was loaded on a HiPrep Q FF column (Cytiva). His-tagged MacV or MvcA was eluted by a 0–500 mM gradient of NaCl in 20 mM Tris–HCl pH7.5, 10 mM 2-mercaptoethanol and was loaded on a HisTrap HP column (Cytiva). His-tagged protein was eluted by a 40–500 mM gradient of imidazole in 20 mM Tris–HCl pH 7.5, 200 mM NaCl, 10 mM 2-mercaptoethanol. Peak fractions were pooled and loaded onto a HiLoad Superdex 200 gel filtration column (Cytiva). Purified protein was eluted in 20 mM Tris–HCl pH 7.5, 200 mM NaCl, 1 mM dithiothreitol and concentrated using a Vivaspin 20 concentrator (Sartorius).

## Ub ligation assay

The in vitro ubiquitination assay in the substrate-free system was conducted as described before (*Kubori et al., 2018*) with minor modifications. Briefly, reaction mixtures (12.5 µl) containing 5 µg of recombinant human Ub (Boston Biochem), 80 nM recombinant human E1 (Boston Biochem), 400 nM recombinant human E2 enzymes (Boston Biochem) and 400 nM purified E3 ligases in 50 mM Tris–Cl (pH 7.5), 2 mM MgCl$_2$, 120 mM NaCl, 2 mM ATP, and 1 mM DTT were incubated for 2 hr at 30°C. The reaction was stopped by adding 12.5 µl of 2× sodium dodecyl sulfate (SDS) sample buffer and boiling.

## Transglutaminase assay

The in vitro transglutaminase assay was conducted using the same buffer for the Ub ligation assay (omitting ATP). Reaction mixtures (12.5 µl) containing 5 µg of Ub, 400 nM purified His-SdcB, and 800 nM purified MavC were incubated for 1 hr at 37°C. The reaction was stopped by adding 12.5 µl of 2× SDS sample buffer and boiling.

## Transfection and infection

HEK293T-FcγRII cells were seeded in poly-L-lysine (Sigma)-coated 6-well tissue culture plates at 6 × 10$^5$ cells/well 24 hr before transfection or infection. Transfection was performed using Lipofectamine 2000 (Invitrogen) for 24 hr according to the manufacturer's recommendation. HeLa-FcγRII cells were seeded on coverslips in 24-well tissue culture plates at 4 × 10$^4$ cells/well 24 hr before transfection or infection. Transfection was performed using polyethylenimine for 24 hr. For infection, *L. pneumophila* was harvested from a 2-day heavy patch grown on CYE plates with or without appropriate antibiotics and 1 mM isopropyl-β-D-thiogalactopyranoside (IPTG), and then it was resuspended in sterilized distilled water. The bacteria were opsonized with anti-*Legionella* antibody (1:3000 dilution) before infection. After adding the bacteria to the cells, the plates were centrifuged at 200 × g to precipitate bacteria onto the layer of cells and were immediately warmed in a 37°C water-bath by floating for 5 min and then placed in a CO$_2$ incubator at 37°C. At 1 hr after infection, the infected cells were washed three times with prewarmed Dulbecco's phosphate-buffered saline (DPBS; Sigma) and refreshed with prewarmed media to remove the extracellular bacteria, and incubation was resumed at 37°C in a CO$_2$ incubator.

## Immunoprecipitation

The transfected or infected cells were washed with DPBS three times and lysed with Lysis buffer (20 mM Tris–HCl, pH 7.5, 150 mM NaCl, 1 mM EDTA, 1% NonidentP40) containing protease inhibitors (cOmplete; Roche), 1 mM phenylmethylsulfonyl fluoride (Nacarai), 10 mM N-ethylmaleimide (NEM, Sigma) as a DUB inhibitor and 10 µM MG132 (Calbiochem) as a proteasome inhibitor. After removal

of cell debris with centrifugation, cell lysates were incubated with FLAG M2 magnetic beads (Sigma), RFP-Trap magnetic beads (chromotek) or Myc-Trap magnetic beads (chromotek) for 2 hr to overnight at 4°C. The beads were washed five times with wash buffer (20 mM Tris–HCl, pH 7.5, 150 mM NaCl, 1 mM EDTA, 0.1% Triton X-100), and the bead-bound proteins were eluted by boiling in SDS sample buffer.

## Immunofluorescent microscopy

HeLa-FcγRII cells on coverslips were fixed with 4% paraformaldehyde/DPBS for 20 min at room temperature and washed with DPBS three times. After permeabilization and blocking with 0.2% Triton X-100 and 2% goat serum in DPBS for 20 min, the coverslips were incubated with the primary antibodies indicated in the figure legends for 90 min. After washing with DPBS three times, the coverslips were incubated with the fluorescent secondary antibodies with 4,6-diamidino-2-phenylindole for 40 min. After washing with DPBS three times, the coverslips were mounted on glass slides using ProLong Diamond antifade mounting medium (Thermo Fisher). Images were collected using an inverted microscope (TE2000-U; Nikon) equipped with a digital ORCA-ERA camera (Hamamatsu).

## Liquid chromatography–MS/MS analysis

Protein bands corresponding to MavC-mediated modification of 3xFLAG-SdcB were excised from SDS–polyacrylamide gel electrophoresis and digested with trypsin. MS experiments were performed at the Research Institute for Microbial Diseases (RIMD). The proteins were reduced with 10 mM DTT, followed by alkylation with 55 mM iodoacetamide, digested by treatment with trypsin (Promega) and purified with a C18 tip (AMR, Tokyo, Japan). The resultant peptides were subjected to nanocapillary reversed-phase liquid chromatography (LC)–MS/MS analysis using a C18 column (12 cm × 75 μm, 1.9 μm, Nikkyo technos, Tokyo, Japan) on a nanoLC system (Bruker Daltoniks, Bremen, Germany) connected to a timsTOF Pro mass spectrometer (Bruker Daltoniks) and a modified nano-electrospray ion source (CaptiveSpray; Bruker Daltoniks). The mobile phase consisted of water containing 0.1% formic acid (solvent A) and acetonitrile containing 0.1% formic acid (solvent B). Linear gradient elution was carried out from 2% to 35% solvent B for 20 min at a flow rate of 250 nl/min. The ion spray voltage was set at 1.6 kV in the positive ion mode. Ions were collected in the trapped ion mobility spectrometry (TIMS) device over 100 ms and MS and MS/MS data were acquired over an $m/z$ range of 100–2000. During the collection of MS/MS data, the TIMS cycle was adjusted to 0.53 s and included 1 MS plus 4 parallel accumulation serial fragmentation (PASEF)-MS/MS scans, each containing on average 12 MS/MS spectra (>100 Hz) (*Meier et al., 2018*; *Meier et al., 2015*) and nitrogen gas was used as collision gas.

The resulting data were processed using DataAnalysis version 5.2 (Bruker Daltoniks), resulting peak files (mgf format) were subjected to MASCOT version 2.7.0 (Matrix Science, London, UK) against the Swissprot_database (568,744 sequences; 205,548,017 residues) taxonomy limited *Homo sapiens* (20,305 sequences), HA-Ub (1 sequences; 92 residues), and 3xFLAG-SdcB database (1 sequences; 950 residues), and searched with the following settings: The mass tolerance for precursor ions was ±15 ppm; The mass tolerance for fragment ions was ±0.05 Da; enzyme, Trypsin; max. missed cleavages, 4; fixed modification: carbamidomethylation on cysteine; variable modifications: oxidation of methionine, N-terminal Gln to pyro-Glu. The threshold score/expectation value for accepting individual spectra was p < 0.05. User defined Crosslinker setting is crosslinker: Ubiq01 (mass modification: −17.026549 Da, deamination). The crosslink reactions were assumed to connect lysine or glutamine. It does not pair with K and K, or Q and Q. It links only 3xFLAG-SdcB between HA-Ub.

## Quantification and statistical analysis

In the immunofluorescence experiments, at least 50 bacterial vacuoles were counted per experiment. Student's *t*-tests were carried out with data from three independent experiments.

## Acknowledgements

We thank Tetsuya Honda, Masanari Nishikawa, Tenne Ichikawa, and Seryeong Du for their technical assistance. LC/MS–MS analysis was conducted by Akinori Ninomiya (Core Instrumentation Facility, Research Institute for Microbial Diseases, Osaka University, Japan). The 3xHA-Ub and 3xHA-Ub-AA expressing plasmids were kind gifts from Jiazhang Qiu (College of Veterinary Medicine, Jilin University,

China). The HA-Ub expressing plasmid is a kind gift from Michinaga Ogawa (National Institute of Infectious Diseases, Japan). We thank Jonathan N Pruneda (Department of Molecular Microbiology & Immunology, Oregon Health & Science University, USA) for providing a critical review of the manuscript. This study was supported by Takeda Science Foundation (to T Ku), MEXT/JSPS KAKENHI grants 22H02867 and 19H03469 (to T Ku), 20H05772 (to K A), 20K07477 (to T Ki), and 19H03470 (to H N).

## Additional information

### Funding

| Funder | Grant reference number | Author |
|---|---|---|
| Takeda Science Foundation | | Tomoko Kubori |
| Japan Society for the Promotion of Science | 22H02867 | Tomoko Kubori |
| Japan Society for the Promotion of Science | 19H03469 | Tomoko Kubori |
| Japan Society for the Promotion of Science | 20H05772 | Kohei Arasaki |
| Japan Society for the Promotion of Science | 20K07477 | Tomoe Kitao |
| Japan Society for the Promotion of Science | 19H03470 | Hiroki Nagai |

The funders had no role in study design, data collection, and interpretation, or the decision to submit the work for publication.

### Author contributions

Tomoko Kubori, Conceptualization, Resources, Data curation, Formal analysis, Supervision, Funding acquisition, Validation, Investigation, Visualization, Writing - original draft, Project administration, Writing - review and editing; Kohei Arasaki, Data curation, Formal analysis, Funding acquisition, Validation, Investigation, Visualization, Writing - review and editing; Hiromu Oide, Data curation, Investigation; Tomoe Kitao, Resources, Funding acquisition, Methodology; Hiroki Nagai, Supervision, Funding acquisition, Investigation, Writing - review and editing

### Author ORCIDs

Tomoko Kubori http://orcid.org/0000-0003-1098-6021
Kohei Arasaki http://orcid.org/0000-0003-0647-3565
Hiroki Nagai http://orcid.org/0000-0003-1659-2197

Reviewer #1 (Public Review): https://doi.org/10.7554/eLife.89002.3.sa1
Reviewer #2 (Public Review): https://doi.org/10.7554/eLife.89002.3.sa2
Author response https://doi.org/10.7554/eLife.89002.3.sa3

## Additional files

### Supplementary files

• Supplementary file 1. Plasmids used in this study are listed.

• Supplementary file 2. Primers used in this study are listed.

• MDAR checklist

### Data availability

Raw data of the LC–MS/MS analysis were deposited in JPOSTrepo (Principal investigator: Tomoko Kubori, Project title: IP-MS analysis of Legionella pneumophila SdcB for detecting noncanonical

ubiquitin crosslink by Legionella transglutaminase MavC, URL to the dataset: https://proteomecentral.proteomexchange.org/cgi/GetDataset?ID=PXD051935. Dataset ID: PXD051935). All data generated or analyzed during this study are included in the manuscript and supplementary files. Source data files have been provided for all figures except the LC–MS/MS data.

The following dataset was generated:

| Author(s) | Year | Dataset title | Dataset URL | Database and Identifier |
|---|---|---|---|---|
| Kubori T | 2024 | IP-MS analysis of Legionella pneumophila SdcB for detecting noncanonical ubiquitin crosslink by Legionella transglutaminase MavC | https://proteomecentral.proteomexchange.org/cgi/GetDataset?ID=PXD051935 | ProteomeXchange, PXD051935 |

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
