## [Editor Report · eLife assessment]

This **important** study explores the interplay between Legionella Dot/Icm effectors that modulate ubiquitination of the host GTPase Rab10, which undergoes phosphoribosyl-ubiquitination by the SidE family of effectors, which in turn are required for Rab10 recruitment to the Legionella containing vacuole (LCV). The evidence supporting the claims of the authors is **convincing**. The study is not only relevant for the microbiology community, but will also be of interest to colleagues in the broader fields of membrane trafficking and general cell biology.

---

## [Referee Report · Reviewer #1 (Public Review)]

This study presents valuable data on effector proteins (=virulence factors) used by the bacterial pathogen Legionella pneumophila that target host vesicle trafficking GTPases during infection. The evidence supporting the claims of the authors is robust, and the data suggest a sophisticated interplay between multiple effectors with the goal of temporarily exploiting host cell Rab10 during infection.

The authors have done a nice job addressing my earlier concerns. I have no further criticism about the revised paper.

---

## [Referee Report · Reviewer #2 (Public Review)]

This manuscript explores the interplay between Legionella Dot/Icm effectors that modulate ubiquitination of the host GTPase Rab10. Rab10 undergoes phosphoribosyl-ubiquitination (PR-Ub) by the SidE family of effectors which is required for its recruitment to the Legionella containing vacuole (LCV). Through a series of elegant experiments using effector gene knockouts, co-transfection studies and careful biochemistry, Kubori et al further demonstrate that:

(1) The SidC family member SdcB contributes to the polyubiquitination (poly-Ub) of Rab10 and its retention at the LCV membrane.

(2) The transglutaminase effector, MavC acts as an inhibitor of SdcB by crosslinking ubiquitin at Gln41 to lysine residues in SdcB.

---

## [Author Response]

The following is the authors’ response to the original reviews.

**eLife assessment**
This study presents valuable findings on Legionella pneumophila effector proteins that target host vesicle trafficking GTPases during infection and more specifically modulate ubiquitination of the host GTPase Rab10. The evidence supporting the claims of the authors is solid, although it remains unclear how modification of the GTPase Rab10 with ubiquitin supports Legionella virulence and the impact of ubiquitination during LCV formation. The work will be of interest to colleagues studying animal pathogens as well as cell biologists in general.

We greatly appreciate the positive and valuable feedback from the editors and the reviewers. According to their suggestions, we added many new experimental data and implications of our findings in Legionella virulence in terms of the biological process of its replication niche. Please find our point-to-point responses below.

**Public Reviews:**

**Reviewer #1 (Public Review):**
In this manuscript, Kubori and colleagues characterized the manipulation of the host cell GTPase Rab10 by several Legionella effector proteins, specifically members of the SidE and SidC family. They show that Rab10 undergoes both conventional ubiquitination and noncanonical phosphoribose-ubiquitination, and that this posttranslational modification contributes to the retention of Rab10 around Legionella vacuoles.StrengthsLegionella is an emerging pathogen of increasing importance, and dissecting its virulence mechanisms allows us to better prevent and treat infections with this organism. How Legionella and related pathogens exploit the function of host cell vesicle transport GTPases of the Rab family is a topic of great interest to the microbial pathogenesis field. This manuscript investigates the molecular processes underlying Rab10 GTPase manipulation by several Legionella effector proteins, most notably members of the SidE and SidC families. The finding that MavC conjugates ubiquitin to SdcB to regulate its function is novel, and sheds further light into the complex network of ubiquitin-related effectors from Lp. The manuscript is well written, and the experiments were performed carefully and examined meticulously.WeaknessesUnfortunately, in its current form this manuscript offers only little additional insight into the role of effector-mediated ubiquitination during Lp infection beyond what has already been published. The enzymatic activities of the SidC and SidE family members were already known prior to this study, as was the importance of Rab10 for optimal Lp virulence. Likewise, it had previously been shown that SidE and SidC family members ubiquitinate various host Rab GTPases, like Rab33 and Rab1. The main contribution of this study is to show that Rab10 is also a substrate of the SidE and SidC family of effectors. What remains unclear is if Rab10 is indeed the main biological target of SdcB (not just 'a' target), and how exactly Rab10 modification with ubiquitin benefits Lp infection.
**Reviewer #1 (Recommendations for The Authors):**
Major points of concern(1) The authors show that SdcB increases Rab10 levels on LCVs at later times of infection and conclude that this is its main biological role. An alternative explanation may be that Rab10 is not 'the main' target of SdcB but merely 'a' target, which may explain why the effect of SdcB on Rab10 accumulation on LCV is only detectable after several hours of infection. An unbiased omics-based approach to identify the actual host target(s) of SdcB may be needed to confirm that Rab10 modification by SdcB is biologically relevant.

We totally agree with your comment that SdcB should have multiple targets considering the abundance of ubiquitin observed on the LCVs when SdcB was expressed (Figure 3). However, the effect of SdcB on Rab10 accumulation at the later time point (7 h) (current Figure 4e) was well supported by the new data showing that the SdcB-mediated ubiquitin conjugation to Rab10 was highly detected at this time point (new Figure 4c). We have tried the comprehensive search of interaction partners of the ANK domain of SdcB. This analysis is planned to be included in our on-going study. We therefore decided not to add the data in this manuscript.

(2) The authors show that Rab10 within cell lysate is ubiquitinated and conclude that ubiquitination of Rab10 is directly responsible for its retention on the LCV. What is the underlying molecular mechanism for this retention? Are GAP proteins prevented from binding and deactivating Rab10. This may be worth testing.

It would be a fantastic hypothesis that a Rab10GAP is involved in the regulation of Rab10 localization on the LCV. However, as far as we know, GAP proteins against Rab10 have not been identified yet. It should be an important issue to be addressed when a Rab10GAP will be found.

(3) Related to this, an alternative explanation would be that Rab10 retention is an indirect effect where inactivators of Rab10, such as host cell GAP proteins, are the main target of SidE/C family members and sent for degradation (see point #1). Can the authors show that Rab10 on the LCV is indeed ubiquitinated?

The possible involvement of a putative Rab10GAP is currently untestable as it is not known. To address whether Rab10 located on the LCV is ubiquitinated nor not, we conducted the critical experiments using active Rab10 (QL) and inactive Rab10 (TN) (new Figure 4a, new Figure 4-figure supplement 1). As revealed for Rab1 (Murata et al., Nature Cell Biol. 2006; Ingmundson et al., Nature 2007), Rab10 is expected to be recruited to the LCV as a GDPbound inactive form and converted to a GTP-bound active form on the LCV. The new results clearly demonstrated that GTP-locked Rab10QL is preferentially ubiquitinated upon infection, strongly supporting the model; Rab10 is ubiquitinated “on the LCV” by the SidE and SidC family ligases.

(4) Also, on what residue(s) is Rab10 ubiquitinated? Jeng et. al. (Cell Host Microbe, 2019, 26(4): 551-563) suggested that K102, K136, and K154 of Rab10 are modified during Lp infection. How does substituting those residues affect the residency of Rab10 on LCVs? Addressing these questions may ultimately help to uncover if the growth defect of a sidE gene cluster deletion strain is due to its inability to ubiquitinate and retain Rab10 on the LCV.

Thank you for the suggestion. We conducted mutagenesis of the three Lys residues of Rab10 and applied the derivative on the ubiquitination analysis (new Figure 1-figure supplement 1). The Lys substitution to Ala residues did not abrogate the ubiquitination upon Lp infection. This result indicates that ubiquitination sites are present in the other residue(s) including the PR-ubiquitination site(s), raising possibility that disruption of sidE genes would be detrimental for intracellular growth of L. pneumophila because of failure of Rab10 retention.

(5) The authors proposed that "the SidE family primarily contributes towards ubiquitination of Rab10". In this case, what is the significance of SdcB-mediated ubiquitination of Rab10 during Lp infection?

We found that the major contribution of SdcB is retention of Rab10 until the late stage of infection. This claim was supported by our new data (new Figure 4c) as mentioned above (response to comment #1).

(6) The contribution of SdcB to ubiquitination of Rab10 relative to SidC and SdcA is unclear. SidC is shown to be unaffected by MavC. In this case, SidC can ubiquitinate Rab10 regardless of the regulatory mechanism of SdcB by MavC. This is not further being examined or discussed in the manuscript.

The effect of intrinsic MavC is apparent at the later stage (9 h) of infection (Figure 7c) when SdcB gains its activity (see above). We therefore do not think that the contribution of MavC on the SidC/SdcA activities, which are effective in the early stage, would impact on Rab10 localization. However, without specific experiments addressing this issue, possible MavC effects on SidC/SdcA would be beyond the scope in this manuscript.

(7) When is Rab10 required during Lp infection? The authors showed that Rab10 levels at LCV are rather stable from 1hr to 7hr post infection. If MavC regulates the activity of SdcB, when does this occur?

While the Rab10 levels on the LCV (~40 %) are stable during 1-7 h post infection (Figure 2b), it reduced to ~20% at 9 h after infection (Figure 7c) (the description was added in lines 304-306). Rab10 seems to be required for optimal LCV biogenesis over the early to late stages, but may not be required at the maturation stage (9 h). We validated the effect of MavC on the Rab10 localization at this time point (Figure 7c). These observations allowed us to build the scheme described in Figure 7d. We revised the illustration in new Figure 7d according to the helpful suggestions from both the reviewers.

(8) Previous analyses by MS showed that ubiquitination of Rab10 in Lp-infected cells decreases over time (from 1 hpi to 8 hpi - Cell Host Microbe, 2019, 26(4): 551-563). How does this align with the findings made here that Rab10 levels on the LCV and likely its ubiquitination levels increase over time?

We carefully compared the Rab10 ubiquitination at 1 h and 7 h after infection (new Figure 1figure supplement 1b). This analysis showed that the level of its ubiquitination decreased over time in agreement with the previous report. Nevertheless, Rab10 was still significantly ubiquitinated at 7 h, which we believe to cause the sustained retention of Rab10 on the LCV at this time point. We added the observation in lines 146-148.

(9) Polyubiquitination of Rab10 was not detected in cells ectopically producing SdcB and SdeA lacking its DUB domain (Figure 7 - figure supplement 2). Does SdcB actually ubiquitinate Rab10 (see also point #5)? Along the same line, it is curious to find that the ubiquitination pattern of Rab10 is not different for LpΔsidC/ΔsdcA compared to LpΔsidC/dsdcA/dsdcB (Figure 1C). The actual contribution of SdcB to ubiquitinating Rab10 compared to SidC/SdcA thus needs to be clarified.

Thank you for the important point. We currently hypothesize that SidC/SdcA/SdcB-mediated ubiquitin conjugation can occur only in the presence of PR-ubiquitin on Rab10 (either directly on the PR-ubiquitin or on other residue(s) of Rab10). Failure to detect the polyubiquitination in the transfection condition (Figure 7-figure supplement 2) suggests that this specific ubiquitin conjugation can occur in the restricted condition, i.e. only “on the LCV”. We added this description in the discussion section (lines 334-335). No difference between the ΔsidCΔsdcA and ΔsidCΔsdcAΔsdcB strains (Figure 1C, 1h infection) can be explained by the result that SdcB gains activity at the later stages (see above).

Minor commentsIn Figure 4b and 7b, the authors show a quantification of "Rab10-positive LCVs/SdcBpositive LCVs". Whys this distinction? It begs the question what the percentile of Rab10positive/SdcB-negative LCVs might be?

We took this way of quantification as we just wanted to see the effect of SdcB on the Rab10 localization. To distinguish between SdcB-positive and negative LCVs, we would need to rely on the blue color signals of DAPI to visualize internal bacteria, which we thought to be technically difficult in this specific analysis.

The band of FLAG-tagged SdcB was not detected by immunoblot using anti-FLAG antibody (Figure 5). The authors hypothesized that "disappearance of the SdcB band can be caused by auto-ubiquitination, as SdcB has an ability to catalyze auto-ubiquitination with a diverse repertoire of E2 enzymes. This can be easily confirmed by using MG-132 to inhibit proteasomal degradation of polyubiquitinated substrates.

We conducted the experiment using MG-132 as suggested and found that proteasomal degradation is not the cause of the disappearance of the band (new Figure 5-figure supplement 2, added description in lines 228-233). SdcB is actually not degraded. Instead, its polyubiquitination causes its apparent loss by distributing the SdcB bands in the gel.

In Figure 5F, the authors mentioned that "HA-UbAA did not conjugate to SdcB", whereas "shifted band detected by FLAG probing plausibly represents conjugation of cellular intrinsic Ub". The same argument was made in Figure 6B. These claims should be confirmed by immunoblot using anti-Ub antibody.

Thank you. We added the data using anti-Ub antibody (P4D1) (Figure 6f, new third panel).

Figure 7A: In cell producing MavC, SdcB is clearly present on LCV. However, in Figure 5A, SdcB was not detected by immunoblot in cells ectopically expressing MavC-C74A. What is the interpretation for these results?

SdcB was not degraded in the cells, but just its apparent molecular weight shift occurred by polyubiquitination (see above). The detection of SdcB in the IF images (Figure 7a) supported this claim.

**Reviewer #2 (Public Review):**
This manuscript explores the interplay between Legionella Dot/Icm effectors that modulate ubiquitination of the host GTPase Rab10. Rab10 undergoes phosphoribosyl-ubiquitination (PR-Ub) by the SidE family of effectors which is required for its recruitment to the Legionella containing vacuole (LCV). Through a series of elegant experiments using effector gene knockouts, co-transfection studies and careful biochemistry, Kubori et al further demonstrate that:(1) The SidC family member SdcB contributes to the polyubiquitination (poly-Ub) of Rab10 and its retention at the LCV membrane.(2) The transglutaminase effector, MavC acts as an inhibitor of SdcB by crosslinking ubiquitin at Gln41 to lysine residues in SdcB.Some further comments and questions are provided below.(1) From the data in Figure 1, it appears that the PR-Ub of Rab10 precedes and in fact is a prerequisite for poly-Ub of Rab10. The authors imply this but there's no explicit statement but isn't this the case?

Yes, we think that it is the case. We revised the description in the text accordingly (lines 326327).

(2) The complex interplay of Legionella effectors and their meta-effectors targeting a single host protein (as shown previously for Rab1) suggests the timing and duration of Rab10 activity on the LCV is tightly regulated. How does the association of Rab10 with the LCV early during infection and then its loss from the LCV at later time points impact LCV biogenesis or stability? This could be clearer in the manuscript and the summary figure does not illustrate this aspect.

Thank you for pointing the important issue. Association of Rab10 with the LCV is thought to be beneficial for L. pneumophila as it is the identified factor which supports bacterial growth in cells (Jeng et al., 2019). We speculate that its loss from the LCV at the later stage of infection would also be beneficial, since the LCV may need to move on to the maturation stage in which a different membrane-fusion process may proceed. As this is too speculative, we gave a simple modification on the part of discussion section (lines 356-358). We also modified the summary figure (revised Figure 7d) as illustrated with the time course.

(3) How do the activities of the SidE and SidC effectors influence the amount of active Rab10 on the LCV (not just its localisation and ubiquitination)

We agree that it is an important point. We tested the active Rab10 (QL) and inactive Rab10 (TN) for their ubiquitination and LCV-localization profiles (new Figure 4ab, new Figure 4figure supplement 1 and 2). These analyses led us to the unexpected finding that the active form of Rab10 is the preferential target of the effector-mediated manipulation. See also our response to Reviewer 1’s comment #3. Thank you very much for your insightful suggestion.

(4) What is the fate of PR-Ub and then poly-Ub Rab10? How does poly-Ub of Rab10 result in its persistence at the LCV membrane rather than its degradation by the proteosome?

We have not revealed the molecular mechanism in this study. We believe that it is an important question to be solved in future. We added the sentence in the discussion section (lines 376378).

(5) Mutation of Lys518, the amino acid in SdcB identified by mass spec as modified by MavC, did not abrogate SdcB Ub-crosslinking, which leaves open the question of how MavC does inhibit SdcB. Is there any evidence of MavC mediated modification to the active site of SdcB?

The active site of SdcB (C57) is required for the modification (Figure 5b), but it is not likely to be the target residue, as the MavC transglutaminase activity restricts the target residues to Lys. It would be expected that multiple Lys residues on SdcB can be modified by MavC to disturb the catalytic activity.

(6) I found it difficult to understand the role of the ubiquitin glycine residues and the transglutaminase activity of MavC on the inhibition of SdcB function. Is structural modelling using Alphafold for example helpful to explain this?

We conducted the Alphafold analysis of SdcB-Ub. Unfortunately, when the Glycine residues of Ub was placed to the catalytic pocket of SdcB, Q41 of Ub did not fit to the expected position of SdcB (K518). Probably, the ternary complex (MavC-Ub-SdcB) would cause the change of their entire conformation. A crystal structure analysis or more detailed molecular modeling would be required to resolve the issue.

(7) Are the lys mutants of SdbB still active in poly-Ub of Rab10?

We performed the experiment and found that K518R K891R mutant of SdcB still has the E3 ligase activity of similar level with the wild-type upon infection (new Figure 6-figure supplement 2) (lines 283-284). The level was actually slightly higher than that of the wildtype. This result may suggest that the blocking of the modification sites can rescue SdcB from MavC-mediated down regulation.

**Reviewer #2 (Recommendations For The Authors):**
see above